# Transcriptional super-enhancers control cancer stemness and metastasis genes in squamous cell carcinoma

Jiaqiang Dong [1,2,6], Jiong Li [3,4,6 ✉], Yang Li[1,2], Zhikun Ma[3,4], Yongxin Yu[1,2] & Cun-Yu Wang [1,2,5 ✉]

Cancer stem cells (CSCs) play a critical role in invasive growth and metastasis of human head and neck squamous cell carcinoma (HNSCC). Although significant progress has been made in understanding the self-renewal and pro-tumorigenic potentials of CSCs, a key challenge remains on how to eliminate CSCs and halt metastasis effectively. Here we show that super-enhancers (SEs) play a critical role in the transcription of cancer stemness genes as well as pro-metastatic genes, thereby controlling their tumorigenic potential and metastasis. Mechanistically, we find that bromodomain-containing protein 4 (BRD4) recruits Mediators and NF-κB p65 to form SEs at cancer stemness genes such as *TP63*, *MET* and *FOSL1*, in addition to oncogenic transcripts. In vivo lineage tracing reveals that disrupting SEs by BET inhibitors potently inhibited CSC self-renewal and eliminated CSCs in addition to elimination of proliferating non-stem tumor cells in a mouse model of HNSCC. Moreover, disrupting SEs also inhibits the invasive growth and lymph node metastasis of human CSCs isolated from human HNSCC. Taken together, our results suggest that targeting SEs may serve as an effective therapy for HNSCC by eliminating CSCs.

[1] Jonsson Comprehensive Cancer Center and Broad Stem Cell Research Center, UCLA, Los Angeles, CA, USA. [2] Laboratory of Molecular Signaling, Division of Oral Biology and Medicine, School of Dentistry, UCLA, Los Angeles, CA, USA. [3] Department of Medicinal Chemistry, School of Pharmacy, Virginia Commonwealth University, Richmond, VA, USA. [4] Department of Oral and Craniofacial Molecular Biology, School of Dentistry, Richmond, VA, USA. [5] Department of Bioengineering, Henry Samueli School of Engineering and Applied Science, UCLA, Los Angeles, CA, USA. [6] These authors contributed equally: Jiaqiang Dong, Jiong Li. ✉email: jli29@vcu.edu; cunywang@ucla.edu

SCC accounts for over 90% of all head and neck malignancies and has a poor prognosis. SCC is highly invasive and frequently metastasizes to cervical lymph nodes[1]. Although SCC initially responds to chemotherapy, patients with SCC rapidly develop chemoresistance and eventually relapse, leading to death. Therefore, novel effective therapies should be developed for HNSCC patients[1]. Recently, using in vivo lineage tracing and genetic approaches, we demonstrated that CSCs plays a critical role in the initiation, metastasis, and chemoresistance of HNSCC. Targeting CSCs in combination with chemotherapy effectively inhibits HNSCC invasive growth and metastasis[2].

Growing evidence suggests that master transcription factors, together with dynamic histone modifications, assemble SEs at cell-type-determining genes to maintain cell identity and status[3,4]. BRD4 is one of the four Bromo- and Extra-Terminal domain (BET) family members, which also includes BRD2, BRD3, and BRDT and are characterized by containing two tandem bromo-domains and one extraterminal domain[5]. BRD4 functions as an epigenetic reader that recognizes and interacts with acetylated lysine residues on histone H3 and H4[6]. Upon binding to the acetylated chromatin locations, BRD4 recruits the Mediator complex, RNA polymerase II (Pol II), as well as the elongation cofactor positive transcription elongation factor b (P-TEFb) to mediate transcription initiation and elongation[7,8]. Notably, recent studies have shown that SEs preferentially regulated the transcription of oncogenes in various cancers, which can be selectively inhibited by BET inhibitors[9,10]. However, most studies involving the use of BET inhibitors to displace BRD4 from acetylated histones were mainly applied to MYC-driving tumors[10–12]. Whether BET inhibitors are also capable of inhibiting human HNSCC growth through suppression of MYC has not been well documented. In the present study, we characterize SEs in SCC, and unexpectedly find that SEs uniquely controlled a set of cancer stemness-associated genes instead of *MYC*. Using both a spontaneous mouse model and human patient-derived xenograft (PDX) model of HNSCC, we show that disrupting SEs by BET inhibitors effectively eliminated BMI1+ CSCs in addition to proliferating non-stem tumor cells, resulting in the inhibition of HNSCC invasive growth and metastasis.

## Results

**BET inhibition suppresses the transcription of cancer stemness genes in HNSCC cells.** To explore whether SEs regulated oncogenic transcription associated with HNSCC, we treated human HNSCC cell lines, including SCC1, SCC22B, and FaDu cells, with the well-known BET inhibitor JQ1 and subsequently performed RNA-seq profiling. RNA-seq results showed that JQ1 treatment inhibited the expression of 1873 genes in SCC1, 1895 genes in SCC22B, and 1892 genes in FaDu, respectively (Fig. 1a and Supplementary Data 1). Unexpectedly, although the oncogene *MYC* has been extensively reported to be a critical target of BET inhibition in various malignancies, it was only mildly inhibited by JQ1 in HNSCC cells (Supplementary Fig. 1a). A geneset enrichment analysis (GSEA) revealed that the expression for nuclear factor-kappa B (NF-κB) target genes was significantly decreased in the JQ1-treated SCC cells compared with control cells (Fig. 1b). NF-κB target genes have been found to be associated with the invasive growth, survival, and metastasis of HNSCC[13–15]. Quantitative reverse transcription-polymerase chain reaction (RT-qPCR) confirmed that JQ1 treatment inhibited the expression of multiple well-known NF-κB target genes including *BCL3, IL1A, IL1B, IL6, IL8, CXCL1, TNFAIP3, BCL2, BIRC3*, and *MMP3* (Fig. 1c), while the expression of NF-κB pathway components, such as RELA, RELB, and IκBα, remained unaffected,

suggesting that JQ1 may directly suppresses NF-κB-dependent transcription (Supplementary Fig. 1b).

Recently, we found that BMI1 was a functional CSC marker and controlled the self-renewal and tumorigenic function of CSCs in HNSCC[2]. We were aware that *BMI1* expression was partially inhibited in SCC cells by JQ1. Therefore, we examined whether JQ1 treatment affected cancer stemness gene signatures. GSEA revealed that the cancer stemness gene signatures were significantly inhibited by JQ1 in all three SCC cell lines (Fig. 1d). qRT-PCR confirmed that JQ1 inhibited the expression of cancer stemness genes, including *TP63, MET, CD44, FOSL1, AURKB, BANF1, BUB1B, CCND2, CDC20, CHAF1A, CKS1B, KIF22, MCM5, NCAPD2*, and *YAP1*[16] (Fig. 1e). Furthermore, we treated SCC1 cells with another BET inhibitor I-BET-151[6], and confirmed that BET inhibition suppressed the expression of NF-κB target genes and cancer stemness genes (Supplementary Fig. 1c). ALDH$^{high}$CD44$^{high}$ are common markers for isolating CSCs from HNSCC cell lines and primary HNSCC tissues. Since cancer stemness genes were inhibited by BET inhibitors, we also directly examined whether JQ1 inhibited the expression of cancer stemness genes in CSCs. qRT-PCR confirmed that JQ1 inhibited the expression of cancer stemness genes and NF-κB target genes in CSCs from human SCC1 cells (Fig. 1f) and human PDXs of HNSCC (Fig. 1g).

BET inhibitors mainly exert their function via disrupting SEs formed by master transcription factors and the Mediator complex at key oncogenes[17]. To explore the dynamic SE alteration during BET inhibition, we performed chromatin immunoprecipitation sequencing (ChIP-seq) of SCC1 cells to examine genome-wide occupancy of the Mediator complex subunit 1 (MED1) and BRD4 following JQ1 treatment. A dramatic decrease of MED1 occupancy was observed in JQ1-treated cells (Fig. 2a). When defined by the enrichment of MED1, 608 SEs were identified in control cells and only 61 were identified in JQ1-treated cells (Fig. 2b). Using anti-H3K27Ac and anti-H3K3me1 antibodies, we identified 569 SEs and 1497 putative SEs, respectively, which included most of MED1-enriched SEs, confirming that MED1-defined SEs have increased H3K27Ac and H3K4me1 (Supplementary Fig. 2a, b). GO analysis indicated that 608 SE-related genes were associated with transcription misregulation in cancer, the NF-κB signaling pathway and PI3K-Akt signaling pathway, indicating that SEs control oncogenic transcription (Fig. 2c).

By overlapping 3648 deferentially expressed genes regulated after JQ1 treatment and 608 SE-associated transcripts, we identified 227 upregulated genes and 193 downregulated genes that are associated with SEs. Studies have shown that key oncogenes driven by SEs are vulnerable to SE disruption. To investigate whether SE-associated genes are disproportionately relying on SEs more than typical enhancers (TEs), we performed the analysis of transcriptional profiles. The results indicated that expression of SE-associated gene were significantly higher than the TE-associated genes (Fig. 2d). In addition, among the genes downregulated by JQ1, the SE-associated genes were more significantly downregulated by JQ1 treatment as compared with those associated with TEs (Fig. 2e), indicating that the high expression of SE-associated genes are relying on SEs.

Among 608 SE-associated transcripts, we found that a total of 41 cancer stemness genes had SEs (Fig. 2b). Unexpectedly, while BMI1 expression was partially inhibited by JQ1, we did not detect SEs in BMI1. Based on the enrichment of BRD4, ChIP-seq identified 678 SEs, which almost completely overlapped with MED1-enriched SEs (Fig. 2f). Although a few SEs could be identified in JQ1-treated cells, the binding intensities were significantly reduced (Fig. 2g). While its genome-wide binding was only moderately inhibited, dramatic BRD4 disruption was

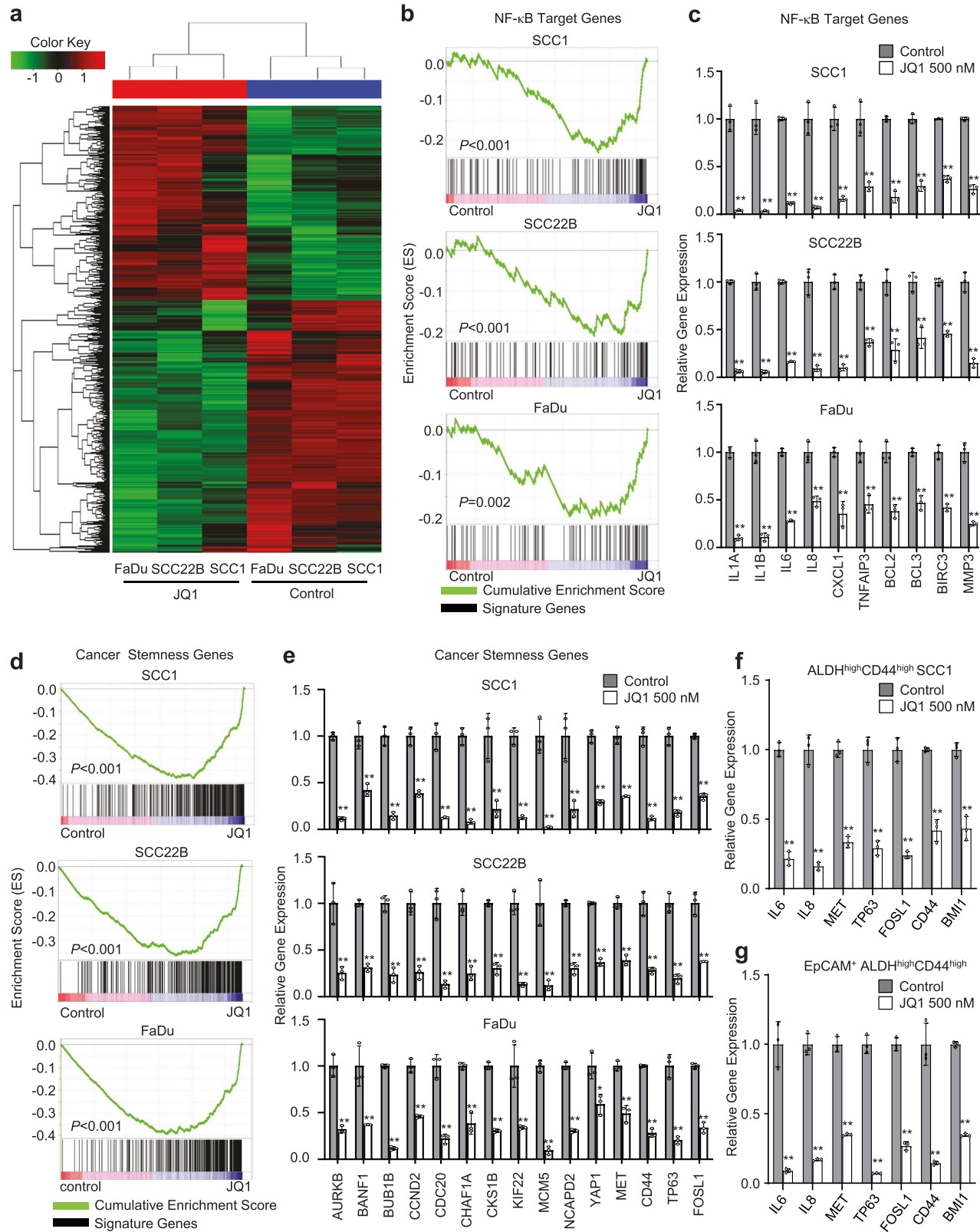

observed in SE regions (Fig. 2g), echoing the fact that SE were more sensitive to BET inhibition.

Due to the fact that JQ1 inhibited the expression of NF-κB target genes, it was possible that NF-κB might be involved in the SE formation in SCC cells[18,19]. p65, also known as RELA, is an active subunit of NF-κB[13,14,20]. Therefore, we profiled p65 chromatin binding with ChIP-seq. p65 displacements in both

genome-wide and SE regions were observed (Fig. 2h). We identified 14935 potential p65-binding sites and 14697 potential BRD4-binding sites, of which, 8022 sites were shared by both p65 and BRD4. After JQ1 treatment, the potential number of p65 and BRD4-binding sites was reduced to 5130 and 11056, respectively. The number of overlapping binding sites was reduced to 2992 following JQ1 treatment. Of the 608 SEs based on MED1

**Fig. 1 BET inhibitors suppress the expression of cancer stemness genes and pro-metastatic genes in HNSCC cells. a** Inhibition of gene expression in FaDu, SCC1, and SCC22B cells by RNA-seq. **b** The expression of NF-κB target genes was significantly decreased by JQ1 as analyzed by GSEA. P-values from the GSEA derived from a permutation test. **c** RT-qPCR showed that JQ1 treatment inhibited the expression of NF-κB target genes ($n = 3$ per group). **d** GSEA identified that JQ1 inhibited cancer stemness genes in SCC cells. P-values from the GSEA derived from a permutation test. **e** RT-qPCR showed that JQ1 inhibited the expression of cancer stemness genes. **f** RT-qPCR showed that JQ1 inhibited the expression of cancer stemness genes and NF-κB target genes in CSC-like cells from human SCC1 cells ($n = 3$ per group). **g** RT-qPCR showed that JQ1 inhibited the expression of cancer stemness genes and NF-κB target genes in CSCs from human PDXs of HNSCC ($n = 3$ per group). Data are presented as mean values ± SD in **c**, **e**, **f**, and **g**. Statistical analysis was performed using two-tailed unpaired Student's t-test. *$P < 0.05$ and **$P < 0.01$. The data in **c**, **e**, **f**, and **g** are representative of three experiments with similar results. Source data are provided as a Source data file. The precise P-values are summarized in Source data file.

enrichment, 605 hold potential binding sites for BRD4, 580 hold potential binding sites for p65, and 570 hold potential binding sites for both p65 and BRD4 (Fig. 2i).

To further explore the molecular mechanisms underlying these events, we first examined the interactions between BRD4, MED1, and p65. An endogenous IP analysis in SCC1 cells confirmed the endogenous interactions between BRD4, MED1, and p65 (Supplementary Fig. 2c). It has been reported that BRD4 co-activates NF-κB-dependent transcription through interaction with acetylated p65 with their BD domain[19]. Thus, we performed an in vitro competition binding assay. GST-BRD4-49-460, which contains both BD1 and BD2 domains, pulled down overexpressed Flag-tagged p65 from 293 T cell lysate. JQ1 inhibited their interaction in a dose dependent manner (Supplementary Fig. 2d). The results imply that JQ1 could release p65 from SEs in addition to the inhibition of BRD4 and histone interaction. Taken together, our studies suggest that that BRD4 selectively co-localizes with MED1 and p65 to form SEs in SCC cells, which can be directly disrupted by BET inhibitors.

**SEs control the expression of the key stem cell gene and pro-metastatic gene.** Although *BMI1* expression was downregulated by JQ1 treatment, our ChIP-seq did not identified any SEs in *BMI1*, indicating that JQ1 treatment might not affect *BMI1* expression through SEs. *TP63* (p63) is a *TP53* (p53) homolog that plays an essential role in the self-renewal and tumorigenic potentials of CSCs[21,22]. Interestingly, ChIP-seq indicated that SEs controlled *TP63* expression. To confirm that *TP63* controlled CSC self-renewal in HNSCC, we isolated ALDH^high^CD44^high CSC-like cells from SCC1 cells and knocked down its expression (Fig. 3a). We found that knockdown of *TP63* abolished tumorsphere formation of CSCs. We also isolated CSCs from human PDXs of HNSCC using EpCAM+ALDH^high^CD44^high markers as previously described[2]. Knockdown of *TP63* also inhibited tumorsphere formation of EpCAM+ALDH^high^CD44^high CSCs (Fig. 3b). The track showed that the signals for MED1, BRD4, and p65 were highly enriched at the SE region of *TP63* that was significantly reduced upon JQ1 treatment (Fig. 3c). To further confirm our ChIP-seq results, we performed ChIP-qPCR and found that JQ1 treatment significantly reduced the occupancy of p65, BRD4 and MED1 on the SE region of *TP63* (Fig. 3d). MET, the receptor of hepatocyte growth factor (HGF), plays a critical role in HNSCC invasive growth and metastasis which has been found to be a functional CSC marker in HNSCC[23,24]. ChIP-seq also identified that MED1, BRD4, and p65 were enriched in the SE region of *MET*. ChIP-qPCR confirmed that JQ1 treatment also significantly diminished the occupancy of BRD4, MED1, and p65 on SEs of *MET* (Fig. 3e, f). BIRC3 and MMP3 promote cancer cell survival and metastasis[20,25–27], respectively. ChIP-qPCR confirmed that BET inhibition significantly disrupted the SEs at *BIRC3* and *MMP3* (Fig. 3g–j).

To evaluate the transcriptional regulatory role of SE loci, we adopted CRISPRa and CRISPRi techniques to recruit either a transcriptional activator or repressor complex to the SE loci with

a catalytically dead Cas9 (dCas9)[28,29]. As shown in Fig. 3k, l, the recruitment of the Synergistic Activation Mediator (SAM) complex to either TP63 or MMP3 SE regions resulted in significant upregulation of *TP63* or *MMP3* expression in SCC1 cells. In comparison, tethered the dCas9-KRAB repressor complex to the same loci significantly inhibited the expression of both genes. To further confirm these findings, we cloned a 2 kb fragment of TP63-SE region as well as a 2 kb negative control region (TP63-NEG) into the pLG4.23 luciferase reporter (Fig. 2m). As compared to the negative control and pLG4.23 promoter, the TP63-SE fragment was capable of elevating the luciferase reporter activity in SCC1 cells (Fig. 2m). Collectively, these results confirmed the functional regulatory role of SE in gene expression. Finally, to further confirm the critical role of SEs in maintaining cancer stemness and promoting metastatic ability, we performed rescue experiments by overexpressing MET in SCC1 cells (Supplementary Fig. 3a). While overexpression of *MET* in SCC1 cells significantly increased tumorsphere formation, the inhibition of tumorsphere formation by JQ1 was less potent in SCC1 cells overexpressing *MET* than in SCC1 cells expressing empty vector (Supplementary Fig. 3b). Similarly, overexpression of MET could also rescue the inhibition of SCC1 cell migration by JQ1 (Supplementary Fig. 3c). Taken together, our data demonstrated that SEs may play a critical role in promoting cancer stemness and metastatic ability of HNSCC through maintaining the high expression of cancer stemness and pro-metastatic genes.

**BRD4 recruits MED1 and p65 to form SEs.** To explore how BRD4, MED1 and p65 form SEs to stimulate SE-target gene transcription, we knocked down them individually (Supplementary Fig. 4a). ChIP-qPCR showed that knockdown of BRD4 led to a significant reduction of both p65 and MED1 enrichments to SEs in *TP63*. Knockdown of p65 significantly inhibited the recruitment of MED1, but not BRD4, to SE in *TP63*. Knockdown of MED1 significantly reduced the recruitment of p65, but not BRD4, to the SE region in *TP63* (Fig. 4a). A co-IP analysis in SCC1 cells also suggested that the interaction between p65 and MED1 is abolished by depletion of BRD4, which further confirmed this finding (Supplementary Fig. 4b). These results revealed that BRD4 plays an important role in the SE formation by recruiting both MED1 and p65 while BRD4 binding in SEs was independent of p65 and MED1. The recruitments of p65 and MED1 to SE were mutually dependent on each other. Similarly, ChIP-qPCR also indicated that BRD4, p65, and MED1 formed SEs in *MET* using a same fashion (Fig. 4b). Since BRD4 and p65 are not only present in SEs but also in promoters and enhancers, we explored whether this mechanism is also applied to these loci other than SEs. We performed similar ChIP-qPCR analysis on the promoter or enhancer regions of well-characterized NF-κB target genes, *CXCL1*, *ICAM1*, and *LTB*, which are not associated with SEs in SCC1 cells. As shown in Supplementary Fig. 4c, knock-down of BRD4 led to a significant inhibition of p65 enrichment to these loci. However, the recruitment of BRD4 remained

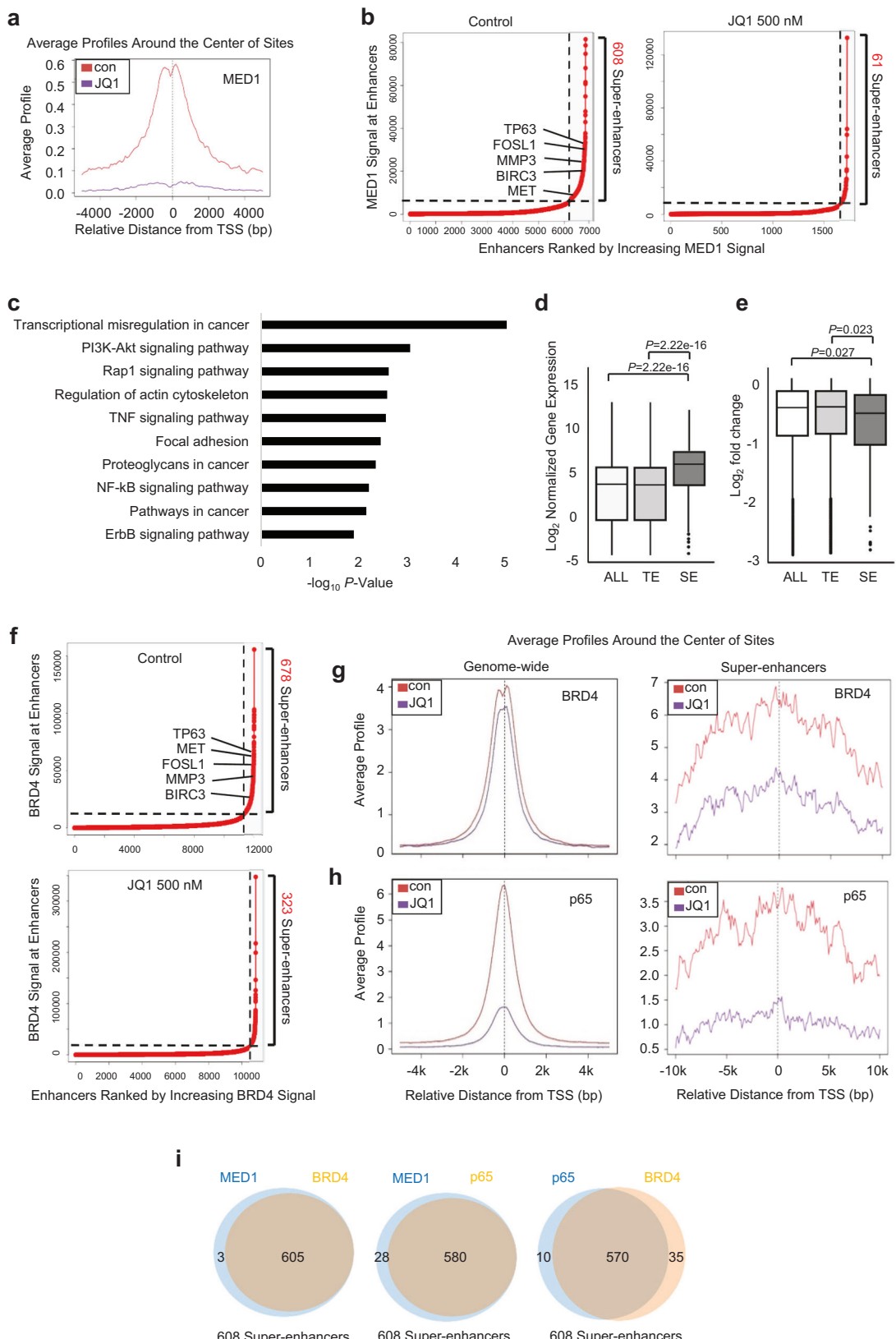

unaffected by depletion of p65. The results suggested that the recruitment of p65 to chromatin by BRD4 is a critical transcriptional mechanism is SCC.

BRD4 has also been reported to recruit the pTEFb (CDK9/CyclinT1) complex[7,8]. This complex plays a critical role in transcriptional elongation[30]. JQ1 treatment significantly inhibited the recruitment of both CDK9 and CCNT1 to SEs in both *TP63* and *MET* (Fig. 4c, d). Moreover, p65 knockdown also diminished the recruitment of both CDK9 and CCNT1 to SEs in both *TP63* and *MET* (Fig. 4c, d). Consistently, RT-qPCR also found that knockdown of p65 or MED1 significantly inhibited the expression of *TP63*, *MET*, *BIRC3*, and *MMP3* (Fig. 4e).

**Fig. 2 SEs control the expression of cancer stemness genes and pro-invasive genes. a** JQ1 inhibited MED1 occupancies on genome by ChIP-seq. **b** JQ1 disrupted SEs in SCC cells based on MED1 enrichments. **c** GO analysis of 608 SE-associated genes in SCC cells. **d** Genes associated with SEs display higher expression than genes associated with total pool of all enhancers (ALL) and typical enhancers (TE) in SCC1 cells. Statistical analysis was performed using two-tailed unpaired Student's *t*-test. Boxplot: middle line of box indicates median and the bounds indicate quartile 1 and quartile 3. **e** Boxplots showing log2 fold changes for downregulated transcripts associated with total pool of ALL, TEs, and SEs after JQ1 treatment in SCC1 cells. Statistical analysis was performed using two-tailed unpaired Student's *t*-test. Boxplot: middle line of box indicates median and the bounds indicate quartile 1 and quartile 3. **f** JQ1 disrupted SEs in SCC cells based on BRD4 enrichments. **g** JQ1 strongly inhibited BRD4 binding in SEs compared with genome-wide regions. **h** JQ1 displaced p65 occupancies in both genome-wide and SE regions. **i** BRD4 selectively colocalized with MED1 and p65 to form SEs in SCC cells.

On the basis of our ChIP-seq and ChIP-qPCR results, we also directly examined the recruitment of BRD4 and p65 on SEs in *TP63* and *MET* in CSCs. As compared with ALDH$^{low}$CD44$^{high}$ non-stem tumor cells, increased enrichments of both p65 and BRD4 on SEs in both *TP63* and *MET* were observed in ALDH$^{high}$CD44$^{high}$ CSCs (Fig. 4f, g). Consistently, RT-qPCR found that ALDH$^{high}$CD44$^{high}$ CSCs had significantly higher *TP63* and *MET* expression levels (Fig. 4h). In addition, we also found that these two genes responded differently to JQ1 treatment between CSCs and non-stem tumor cells. JQ1 had little effect on *TP63* and *MET* expression in ALDH$^{low}$CD44$^{high}$ non-stem tumor cells (Supplementary Fig. 4e).

**BET inhibitors eliminate CSCs in vivo and prevent lymph node metastasis.** Our results suggest that SEs not only control pro-oncogenic gene transcription, but also genes associated with CSC self-renewal and function. Based on these findings, we hypothesized that BET inhibition might help to eliminate CSCs in HNSCC in addition to non-stem tumor cells. Previously, we have established a 4NQO-induced HNSCC Bmi1$^{CreER}$;Rosa$^{tdTomato}$ mouse model which allows us to perform the lineage tracing of CSCs in an in vivo unperturbed environment using tamoxifen-induced Cre-mediated recombination[2]. To test our hypothesis, we examined whether JQ1 could inhibit SCC invasive growth and metastasis by eliminating CSCs. Mice were treated with 4NQO in their drinking water for 16 weeks. At 22 weeks from the initial 4NQO treatment, the tumor-bearing mice were administered with JQ1 or vehicle control five times a week for 4 weeks (Fig. 5a). Histological examination found that JQ1 significantly inhibited squamous epithelial dysplasia (Fig. 5b). Because each mouse developed several tumors upon 4NQO treatment, we counted all tumors from each group based on histological analysis. Histological evaluation demonstrated that JQ1 treatment significantly induced the regression of overt SCC (Fig. 5c). More importantly, anti-pan cytokeratin (PCK) staining revealed that very fewer cervical lymph nodes contained metastatic lesions in mice that were treated with JQ1 (Fig. 5d, e). A single dose of tamoxifen was also administered prior to the JQ1 treatment to induce Cre-mediated recombination (Fig. 5a). The lineage tracing showed that both PCK$^+$Tomato$^+$ (Bmi1$^+$ CSCs) and PCK$^+$Tomato$^-$ tumor cells were eliminated in the regressed SCCs (Fig. 5f). These data indicate that JQ1 treatment, unlikely chemotherapeutic drugs, targeted Tomato$^+$ CSCs in addition to proliferating non-stem tumor cells by disrupting SEs. To further confirm our results, six successive JQ1 doses were applied to the tumor-bearing mice (Fig. 5g). To label the BMI1$^+$ CSCs, a single dose of tamoxifen was intraperitoneally injected a day before euthanasia. The mitotic cells were stained with the anti-Ki67 antibody in combination with a cy2-conjugated secondary antibody. We confirmed that JQ1 treatment also eliminated Tomato$^+$ CSCs in addition to non-stem Ki67$^+$ proliferating tumor cells (Fig. 5h).

**Disrupting SEs suppresses the tumorigenic potential and metastasis of human CSCs.** To further examine whether BET

inhibition eliminates CSCs, we measured the proportion of the CD44$^{high}$ALDH$^{high}$ cells in human SCC1 and SCC22B cells upon JQ1 treatment. FACS analysis revealed that JQ1 treatment significantly reduced CD44$^{high}$ALDH$^{high}$ CSC-like cells in SCC1 (Fig. 6a) and SCC22B (Supplementary Fig. 5a). JQ1 treatment also significantly inhibited the formation of tumorspheres mediated by CD44$^{high}$ALDH$^{high}$ CSC-like cells (Fig. 6b, c; Supplementary Fig. 5b). To further confirm our results, we also isolated EpCAM$^+$ CD44$^{high}$ALDH$^{high}$ cells from human PDXs of HNSCC (Fig. 6d). JQ1 treatment also significantly inhibited the tumorsphere formation of EpCAM$^+$CD44$^{high}$ALDH$^{high}$ CSCs (Fig. 6e, f). Moreover, another BET inhibitor, I-BET-151, also suppressed the sphere-forming potential of CSCs isolated from SCC1 cells or human PDXs of HNSCC (Supplementary Fig. 5c, d).

Previously, we demonstrated that cisplatin, a commonly used chemotherapeutic drug for HNSCC, enriches CSCs and promotes metastasis[1]. This was further validated by flow cytometry analysis, which showed that the cisplatin-resistant SCC1 (SCC1-cis) cells displayed higher CD44$^{high}$ALDH$^{high}$ CSC-like proportion. This proportion was remarkably reduced by JQ1 treatment (Fig. 6g). We further examined whether JQ1 could overcome cisplatin resistance and inhibit tumor growth and metastasis in vivo using an orthotopic model of HNSCC[2,27]. We isolated CD44$^{high}$ALDH$^{high}$ CSC-like cells from SCC1-cis cells and sublingually inoculated them into the mouse tongue. JQ1 treatment was able to overcome cisplatin resistance and significantly inhibited tumor growth in vivo (Fig. 6h). To accurately determine lymph node metastasis, we harvested all cervical lymph nodes from each group and immunostained metastatic tumor cells in lymph nodes using anti-PCK antibodies. Immunostaining revealed that there were significantly less lymph nodes that contained metastatic lesions in mice that were treated with JQ1 as opposed to vehicle control (Fig. 6i,j).

To further confirm our results, we also isolated EpCAM$^+$CD44$^{high}$ALDH$^{high}$ CSCs from human PDXs of HNSCC and sublingually inoculated these CSCs into mouse tongues. The mice were then treated with either JQ1 or vehicle control. Histological analysis revealed that the orthotopic tumor formation rate was much higher in the control mice (8 out of 9) than in JQ1-treated mice (3 out of 8). Moreover, the control mice formed larger tumors than JQ1-treated mice (Fig. 6k). We also isolated all cervical lymph nodes from each group and compared their metastasis. Immunostaining with anti-pan cytokeratin found that JQ1 significantly inhibited lymph node metastasis of human CSCs (Fig. 6l, m).

**Discussion**
Our study showed that SEs play a critical role in control of CSC self-renewal and pro-tumorigenic potentials. Traditional chemotherapeutics, such as cisplatin, are effective at abolishing the actively proliferating tumor cells, but are usually not effective for eradicating CSCs, which leads to CSC enrichment and tumor recurrence[31–34]. Importantly, we showed that BET inhibitors potently eliminated both CSCs and rapidly proliferating cancer

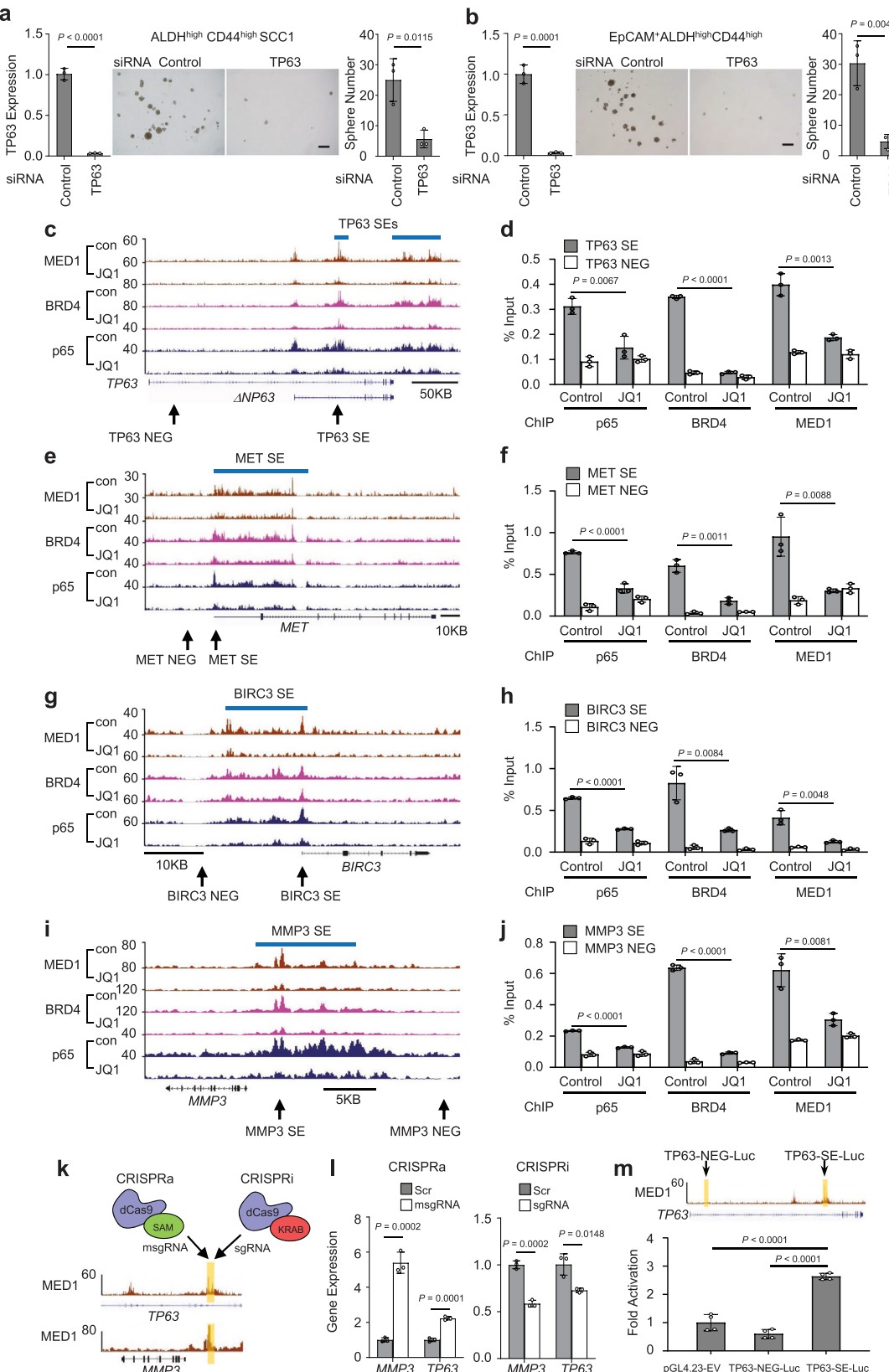

cells by disrupting SEs using in vivo lineage tracing. Mechanistically, we found that SEs potently controlled the expression of cancer stemness genes and pro-metastatic genes, and disrupting SEs by BET inhibitors significantly inhibited their transcription.

Constitutive NF-κB activation is a key feature of HNSCC and is responsible for acquisition of most hallmarks of HNSCC including uncontrolled proliferation, invasion and metastasis, and angiogenesis[13–15]. Since IκBα kinase-β (IKKβ) plays a critical role in NF-kB activation, IKKβ is considered as an important target for HNSCC. However, targeting IKKβ might have broad side-effects given its broad roles in human immunity and host responses. Interestingly, our RNA-seq and ChIP-seq results

**Fig. 3 SEs control the expression of cancer stemness genes and pro-metastatic genes. a** Knockdown of *TP63* inhibited tumorsphere formation by ALDH[high]CD44[high] CSC-like cells from SCC1 cells. Values are mean ± SD for triplicate experiments. Statistical analysis was performed using two-tailed unpaired Student's *t*-test. Scale bar, 200 μm. **b** Knockdown of *TP63* inhibited tumorsphere formation by CSCs from the HNSCC PDXs. Values are mean ± SD for triplicate experiments. Statistical analysis was performed using two-tailed unpaired Student's *t*-test. Scale bar, 200 μm. **c** The signals for MED1, BRD4, and p65 were highly enriched at the SE region of *TP63*. **d** ChIP-qPCR showed that JQ1 treatment significantly reduced the occupancy of p65, BRD4, and MED1 on the SE region of *TP63* ($n = 3$ per group). **e** The signals for MED1, BRD4, and p65 were highly enriched at the SE region of *MET*. **f** ChIP-qPCR showed that JQ1 treatment significantly reduced the occupancy of p65, BRD4, and MED1 on the SE region of *MET* ($n = 3$ per group). **g** The signals for MED1, BRD4, and p65 were highly enriched at the SE region of *BIRC3*. **h** ChIP-qPCR showed that JQ1 treatment significantly reduced the occupancy of p65, BRD4, and MED1 on the SE region of *BIRC3* ($n = 3$ per group). **i** The signals for MED1, BRD4, and p65 were highly enriched at the SE region of *MMP3*. **j** ChIP-qPCR showed that JQ1 treatment significantly reduced the occupancy of p65, BRD4, and MED1 on the SE region of *MMP3* ($n = 3$ per group). **k** Schematic showing targeting SEs using a catalytically dead Cas9 (dCas9) associated with either a transcriptional activator (SAM) or repressor complex (KRAB-MeCP2). **l** RT-qPCR showed that *MMP3* and *TP63* expression was significantly enhanced or inhibited in SCC1 cell following introducing the dCas9-SAM or dCas9-KRAB system targeting SE regions ($n = 3$ per group). **m** TP63-SE and TP63-NEG activities were measured by luciferase reporter assays ($n = 4$ per group). Data are presented as mean values ± SD in **d**, **f**, **h**, **j**, **l**, and **m**. Statistical analysis was performed using two-tailed unpaired Student's *t*-test. The data are representative of three experiments with similar results. Source data are provided as a Source data file.

revealed that BRD4 preferentially recruits p65 and MED1 to form SEs in HNSCC. Our results suggest that p65-associated SEs are also vital targets for HNSCC instead of IKKβ. More importantly, consistent with the reduced self-renewal and metastatic capabilities of HNSCC cells, SEs disruption at corresponding target genes was observed. Besides the cytokines like IL6 and IL8, that exert wide spectrum effect on the biological behavior of tumor cells, we found that disrupting SEs by BET inhibitors inhibited a set of cancer stemness genes, including *TP63*, *MET*, *FOSL1*, and *YAP1*. *TP63* is an isoform of the *p63* family and plays an important role in the maintenance of the epithelial stem repertoire. It was reported to be essential in keeping mammary gland stem cell activity and promoting oncogenesis of basal-like breast cancer[22]. In lung and esophageal SCC, *TP63* was reported to be colocalized with SOX2, another important transcription factor regulating CSC properties[21]. In our study, we demonstrated that BET inhibition downregulated TP63 expression. Silencing *TP63* was also found to suppress the self-renewal of CSC, suggesting TP63 is a bona fide BET target that also participates in maintaining the CSC populations in HNSCC. We and others have demonstrated that *MET*, *FOSL1*, and *YAP1* regulate CSC self-renewal and promoted HNSCC metastasis[2,23,24], further indicating that targeting SEs helps eliminate CSCs.

At present, cisplatin and platinum-based chemotherapy is still the first-line treatment of many solid tumors, especially for HNSCC patients. However, resistance to cisplatin has become a major obstacle for effective therapy of HNSCC. It is well-known that cisplatin is capable of enriching the CSC population, which ultimately contributes to chemoresistance. Our in vitro data suggests that BET inhibition is able to suppress cisplatin-enriched CSC population. More importantly, the in vivo growth of cisplatin-resistant HNSCC cells can also be strongly suppressed by BET inhibition. Our previous study suggested that increased FOSL1 activity facilitates the metastasis and cisplatin resistance of Bmi1[+]CSC in a spontaneous mouse model of HNSCC[2]. Coinciding with this, upregulation of both *BMI1* and *FOSL1* was found in cisplatin-resistant SCC cells. Of note, this upregulation can be significantly suppressed by JQ1. This indicates that JQ1 may overcome cisplatin resistance through inhibition of *BMI1* and *FOSL1* expression, which may represent the molecular mechanism underlying this event. This evidence indicates that BET inhibition may be an effective therapeutic approach to overcome drug resistance in cancer therapy. Thus, future studies will be focused on exploration and validation of whether BET inhibition can effectively eliminate cisplatin-enriched CSCs in HNSCC. Moreover, our in vivo data also demonstrated that JQ1 has the ability to eliminate non-stem tumor cells as well. In agreement with this, we found that besides cancer stemness genes,

the key oncogenes associated with cell proliferation, such as *CCND1*, can also be significantly suppressed by JQ1, which supports the phenomena that JQ1 targets both quiescent CSCs and fast proliferating non-stem tumor cells. To date, several BET inhibitors, such as I-BET-762 (Molibresib), PLX51107, and ABBV-075, have already reached to phase I/II clinical trials for the treatment of solid tumors, including HNSCC[35–38]. Thus, it is anticipated that BET inhibition may be adopted as an effective strategy for future HNSCC therapy. In conclusion, our results suggest that disrupting SEs by BET inhibitors is an effective approach to suppress the growth and metastasis of HNSCC by eliminating CSCs and the mitotic bulk tumor simultaneously.

## Methods

**Cell lines**. Human HNSCC cell lines SCC1 and SCC22B were kindly provided by Dr. Tom Carey at the University of Michigan. FaDu cells were purchased from ATCC (HTB43). These cells were maintained in DMEM containing 10% FBS and antibiotics (streptomycin and penicillin) at 37 °C in a humidified 5% $CO_2$ atmosphere as described before[27]. The cisplatin-resistant SCC1-cis cell line was established by chronic culture in the gradually increased concentration of cisplatin and maintained in 5 μM of cisplatin. For siRNA transfection, HNSCC cells were plated at 40–50% confluence and incubated overnight. For transient knockdown, sequence specific siRNA or scramble control siRNA (Sigma–Aldrich) was transfected using Lipofectamine RNAiMAX Transfection Reagent (Thermo Fisher Scientific, Cat#13778150) according to the manufacturer's manual. Forty-eight hours post transfection, the cells were harvested for RT-qPCR or ChIP analysis.

**4NQO mouse model, lineage tracing, and JQ1 treatment**. All experimental mice were under the husbandry care of the Division of Laboratory Animal Medicine at UCLA. The animal facility was maintained at under a 12 h reversed light-dark cycle with controlled temperature (20–26 °C) and humidity (30–70%). Food and water were available ad libitum. Bmi1[CreER] (Jackson Lab, cat#010531), R26[tdTomato] (Jackson Lab, cat#007908), and NOD/SCID (Jackson Lab, cat#001303) mouse strains were purchased from The Jackson Laboratory. Bmi1[CreER];R26[tdTomato] mice were generated and characterized as previously described[2]. Nude mice were purchased from Taconic (cat#NCRNU-F). Mice were housed under standard conditions in the UCLA animal facility. All procedures were performed according to the protocol #2007-062-41 approved by the UCLA Animal Research Committee. For HNSCC induction, four-week-old C57BL/6 mice were subjected to drinking water containing 50 μg/ml 4NQO (Sigma–Aldrich, cat#N8141) for 16 weeks and then turned to normal drinking water for another 6 weeks. For lineage tracing, Cre was activated by intraperitoneal injection of tamoxifen (Sigma–Aldrich Cat#T5648) at a dose of 9 mg per 40 g body weight as described before[2]. For the treatment, the animals were randomly assigned into either vehicle control or JQ1 treatment group. JQ1 (Tocris, cat#4499) stock solution was prepared in dimethylsulfoxide (Sigma–Aldrich, cat#D2650) at 200 mg/ml. It was further diluted to 5 mg/ml in PBS containing 10% (v/v) Kolliphor® EL (Sigma–Aldrich, cat#C5135) and administered intraperitoneally at 50 mg/kg body weight.

For treatment evaluation, tongues and cervical lymph nodes were dissected and harvested immediately after mice were euthanized. Longitudinally cut tongues (dorsal/ventral) and intact lymph nodes were fixed overnight in 10% PBS buffered formalin, embedded in paraffin and 5 μm sections were made. Dysplasia was graded according to the architectural changes of the epithelium: mild (grade 1), only the lower 1/3 was affected, moderate (grade 2), lower 2/3 was affected, and severe (grade 3), greater than 2/3 of the epithelium architecture was affected. The

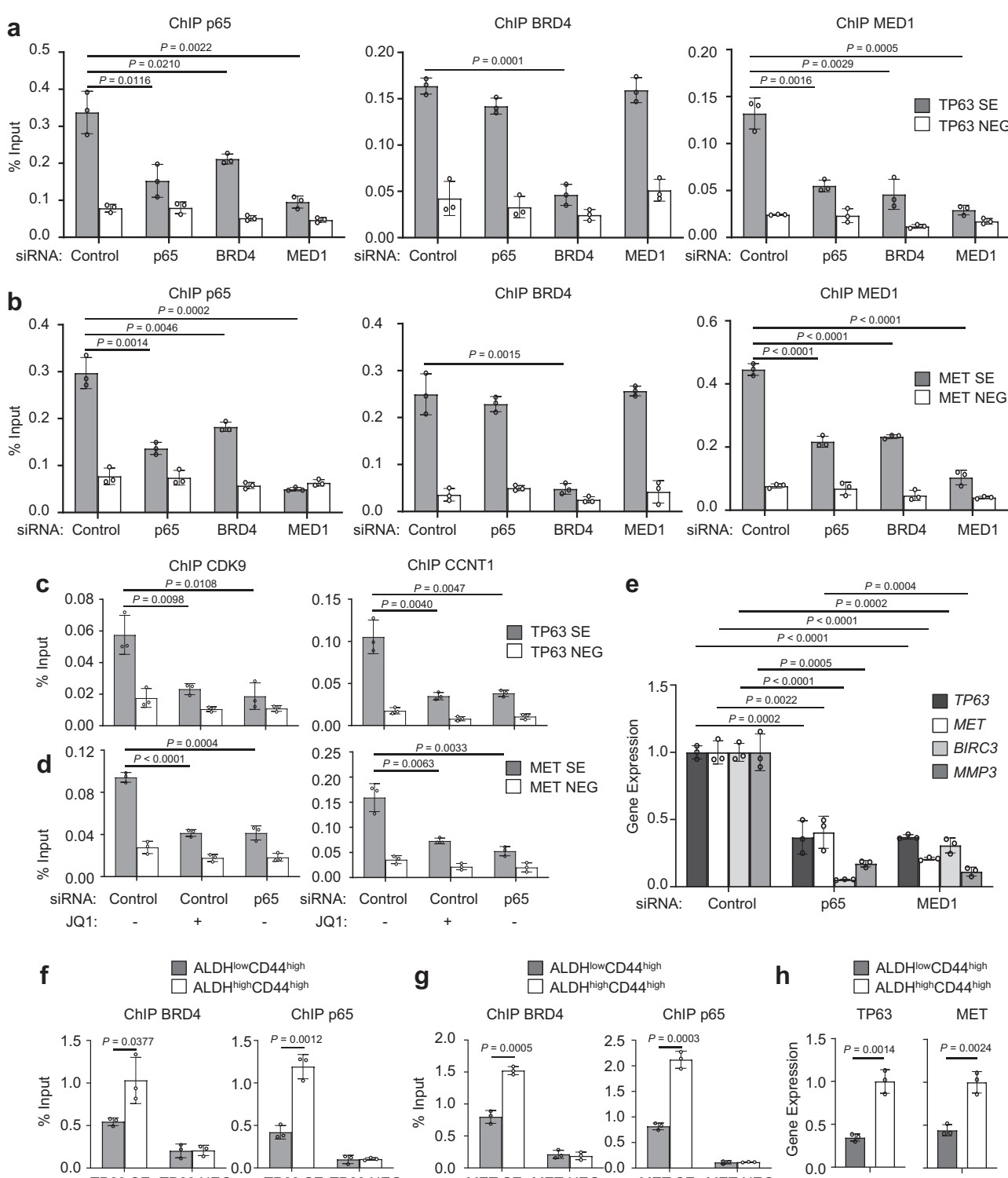

**Fig. 4 BRD4 recruits MED1 and p65 to form SEs. a** The knockdown of BRD4 impaired both p65 and MED1 recruitments to SEs in *TP63* (*n* = 3 per group). **b** The knockdown of BRD4 impaired both p65 and MED1 recruitments to SEs in *MET* (*n* = 3 per group). **c** JQ1 treatment or p65 knockdown significantly inhibited the recruitment of both CDK9 and CCNT1 to SEs in *TP63* (*n* = 3 per group). **d** JQ1 treatment or p65 knockdown significantly inhibited the recruitment of both CDK9 and CCNT1 to SEs in *MET* (*n* = 3 per group). **e** RT-qPCR showed that knockdown of p65 and MED1 significantly inhibited the expression of *TP63*, *MET*, *BIRC3*, and *MMP3* (*n* = 3 per group). **f** Increased enrichments of both p65 and BRD4 on SEs of *TP63* in CSC-like cell (*n* = 3 per group). **g** Increased enrichments of both p65 and BRD4 on SEs of *MET* in CSC-like cells (*n* = 3 per group). **h** CSCs had significantly higher *TP63* and *MET* expression levels (*n* = 3 per group). Data are presented as mean values ± SD in **a–m**. Statistical analysis was performed using two-tailed unpaired Student's *t*-test. The data in **a–m** are representative of three experiments with similar results. Source data are provided as a Source data file.

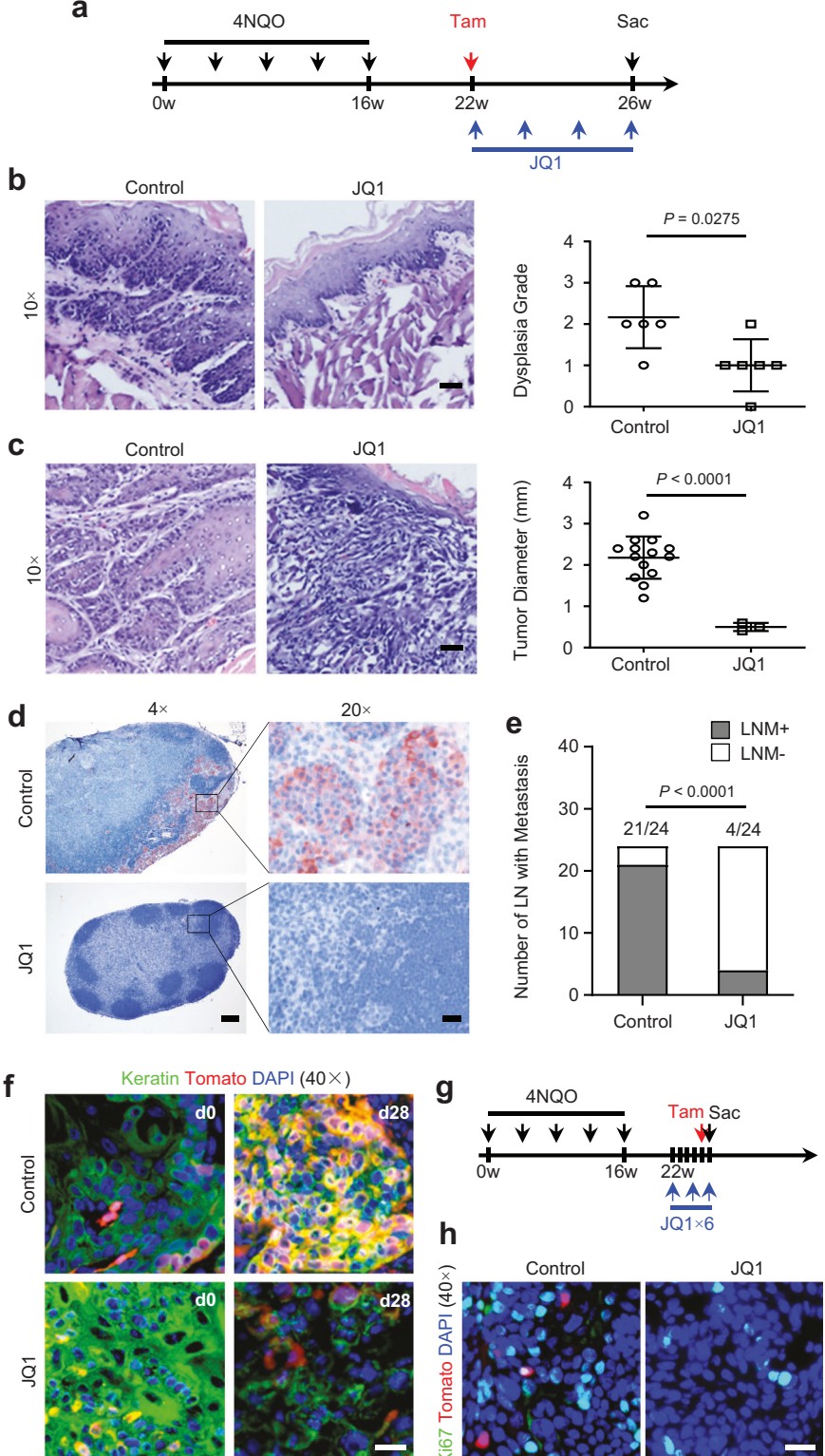

size of SCC was indicated by the diameter. The sections of cervical lymph nodes were immunostained with anti-pan cytokeratin antibodies (Santa Cruz, cat#sc-8018, 1:500). The percentage of lymph nodes with metastasis was calculated.

**Immunostaining**. Mouse HNSCC tumors were fixed with 4% paraformaldehyde overnight, then equilibrated in 30% sucrose, and embedded in OCT (Tissue Tek, cat#25608-930). Eight-micrometer-thick sections were cut using a Leica cryostat at −20 °C and placed on Superfrost Slides (Fisher Scientific Cat#12-550-15). The sections were stained with anti-pan-cytokeratin (Santa Cruz, cat#sc-8018, 1:500) or anti-Ki67 primary antibodies (Abcam, cat#ab15580; 1:400). The signals were

detected using secondary antibodies conjugated with Cy2 (Jackson ImmunoResearch Laboratories). Sections were counterstained with 4'6'-diamidino-2-phenilindole (DAPI; Sigma–Aldrich, cat#D9542), mounted with SlowFade Antifade Reagents (Thermo Fisher Scientific, cat#S36937), and examined under a fluorescent microscopy.

**Human PDXs of HNSCC**. The use of anonymized human remnant HNSCC samples for this study from the UCLA Translational Pathological Core Laboratory (TPCL) was approved by the UCLA Institutional Review Board (IRB#11-002504). Patients signed the informed consent under this IRB approval. The human HNSCC

**Fig. 5 BET inhibitors potently eliminate CSCs in vivo and prevent lymph node metastasis in 4NQO-induced mouse model of HNSCC. a** Experimental plan used to trace Bmi1[+] cells in 4NQO-induced mouse HNSCC upon JQ1 treatment. **b** JQ1 prevented squamous epithelial dysplasia ($n = 6$ per group). Data are presented as mean values ± SD. Statistical analysis was performed using Mann–Whitney test. Scale bar, 100 μm. **c** JQ1 treatment inhibited HNSCC formation ($n = 6$ per group). Data are presented as mean values ± SD. Statistical analysis was performed using two-tailed unpaired Student's *t*-test. Scale bar, 100 μm. **d** Immunostaining of metastatic cells in cervical lymph nodes using anti-pan-cytokeratin (PCK) ($n = 6$ per group; 24 lymph nodes were retrieved from each group). Scale bar left, 250 μm; Scale bar right, 50 μm. **e** Number of cervical lymph nodes with metastasis (LNM) in different groups ($n = 4$ per group; 24 lymph nodes were retrieved from each group). Statistical analysis was performed using two-tailed Fisher exact test. **f** In vivo lineage tracing of BMI1[+] cells (Red) in mouse HNSCC upon JQ1 treatment ($n = 6$ per group). Scale bar, 50 μm. **g** Experimental design used to label Bmi1[+] CSCs after JQ1 treatment in mouse HNSCC. **h** Immunostaining of Ki67 (green) and Bmi1[+] CSCs (red) in mouse HNSCC upon JQ1 treatment ($n = 6$ per group). Scale bar, 50 μm. The experiments in **b, c, d, e, f,** and **h** were repeated once with similar results.

primary tissues from the TPCL were subcutaneously inoculated into flank of 6-week-old NOD/SCID mice to generate the PDXs of HNSCC as described previously[2]. To isolate CSCs from HNSCC PDXs, mice were euthanized and tumors were dissected and isolated. Tumors were chopped into small pieces with scalpels in petri dishes on ice, and then dissociated into single cell suspension using Tumor Dissociation Kit (Miltenyi, cat#130-095-929). The cell suspension was filtered through a 100-μm mesh filter and then followed by a 40-μm mesh filter. Red blood cells were removed using a RBC Lysis Buffer (Santa Cruz Biotechnology, cat#sc-296258). Subsequently, tumor cells were stained with the ALDHEFLUOR assay kit (STEMCELL Technologies Cat#01700) following the manufacturer's guidelines to label the ALDH[high] populations. Tumor cells were then incubated with anti-EpCAM-PE (Miltenyi, cat#130-111-116, 1:100) and anti-CD44-APC (Miltenyi, cat#130-113-893, 1:100) for 30 min at room temperature. Gating strategy was shown in Supplementary Fig. 6. EpCAM[+]ALDH[high]CD44[high] CSCs were sorted by a FACSVantage SE (Beckton Dickson) as described previously (2). The tumorigenic potentials and self-renewal of EpCAM[+]ALDH[high]CD44[high] CSCs were characterized at our lab as previously described[2,27]. To isolate CSC-like cells from HNSCC cell lines, cells were stained with the ALDHEFLUOR assay kit, followed by incubation with anti-CD44-APC. Sorting results were analyzed with BD FACSDiva Version 6 software. For orthotopic nude mouse model of HNSCC, isolated EpCAM[+]ALDH[high]CD44[high] CSCs ($1 \times 10^4$ cells) were mixed with Matrigel and injected sublingually into nude mice. The tumor growth was monitored every day and allowed to grow 2–3 weeks.

**Tumorsphere formation assays**. For tumorsphere formation assays, FACS-sorted cells were cultured in ultralow attachment plates at a density of 100–500 cells/well as described before[32]. Briefly, cells were cultured in serum-free DMEM/F12 (Thermo Fisher Scientific, cat#11330-032) supplemented with 1% B27 supplement (Thermo Fisher Scientific, cat#17504044), 1% N2 supplement (Thermo Fisher Scientific, cat#17502048), penicillin-streptomycin (100 μg/ml; Thermo Fisher Scientific, cat#15140122), human recombinant epidermal growth factor (EGF; 20 ng/ml; R&D Systems, cat#236-EG-01M), and human recombinant basic fibroblast growth factor (bFGF; 10 ng/ml; R&D Systems, cat#233-FB-025/CF), in a humidified 5% $CO_2$ incubator at 37 °C. Spheres with a diameter over 40 μm were counted 1–2 weeks after plating.

**IP and GST pull-down assays**. For each IP sample, ~5 million of SCC1 cells were lysed with CelLytic M buffer (Sigma, cat#C2978) for 5 min on ice. The total cell lysate was incubated with indicated antibody or IgG at 4 °C overnight, and followed by incubation with Protein G Dynabeads (Thermo Fisher, cat#10006D) for 2 h. Immunoprecipitates were washed with PBS plus 0.1% NP-40 buffer for at least three times. To obtain the Flag-tagged p65, 293 T cells were transfected with Flag-tagged p65 expression vector. Two days post transfection, the cells were lysed with CelLytic M buffer. For GST pull-down assays, 2 μg of GST or GST fusion BRD4-49-460 proteins were incubated with glutathione beads (Pierce, cat#16100) at 4 °C for 2 hr. After washing with PBS, beads were then incubated with total cell lysate of 293 T containing Flag-tagged p65 and increasing concentration of JQ1 as indicated for additional 2 h. The beads were with PBS plus 0.1% NP-40 buffer for three times. Proteins bound to the beads were eluted with 1 × SDS-loading buffer at 98 °C for 5 min and then subjected to SDS-PAGE and western blot analysis.

**Western blotting**. Cells were lysed using the CelLytic buffer (Sigma–Aldrich, cat#C3228). Protein extracts were separated on SDS-PAGE before being transferred to a PVDF membrane. Membranes were blocked with 5% milk for 1 h and incubated with primary antibodies overnight, followed by incubation with the secondary antibodies for 2 h at room temperature. Primary antibodies used in this study were: anti-p65 (Sigma; cat#17-10060; 1:5000); anti-MED1 (Bethyl, cat#A300-793A; 1:1000); anti-BRD4 (Abcam, cat#ab128874; 1:5000); anti-α-Tubulin (Sigma, cat#T5168; 1:50000); Goat Anti-Mouse IgG Antibody, HRP conjugate (Sigma, cat#12-349; 1:5000); and anti-Rabbit IgG (H + L), HRP Conjugate (Promega, cat#W4011; 1:7500). Full scans of western blots were provided in the Source Data file.

**RNA-seq, geneset enrichment analysis, and RT-qPCR**. Quality of the RNA for sequencing was determined using an Agilent 2100 Bioanalyzer. Library preparation using the KAPA RNA-Seq Library Preparation Kits (KAPA Biosystems, cat#07960140001) was performed at the UCLA sequencing core facility, and RNAs were single-end sequenced on Illumina HiSeq 3000 machines. Reads were aligned using Tophat with the hg18 build of the human genome (https://ccb.jhu.edu/software/tophat/index.shtml). Transcript assembly and differential expression were determined using Cufflinks with Refseq mRNAs. RNA-seq data were analyzed using the cummeRbund package in R (http://cole-trapnell-lab.github.io/cufflinks/). The heatmap was generated with ComplexHeatmap Version 2.6.2 (http://ashleylab.stanford.edu/tools/tools-scripts.html). Transcripts regulated both greater than and less than two-fold were used in GO term and KEGG pathway analysis to detect enriched functional annotations. GSEA and the statistical analyses were performed with GSEA software Version 3.0 (http://www.broad.mit.edu/GSEA) and a two-tailed *t*-test, respectively.

For RT-qPCR, total RNA was extracted using TRIzol reagent (Thermo Fisher Scientific, cat#15596026), and 1 μg of RNA was used for the RT reaction with random primer (Thermo Fisher Scientific, cat#48190011), dNTP mix (Thermo Fisher Scientific, cat#18427013), and M-MuLV Reverse Transcriptase (New England Biolabs, cat#M0253L). The transcripts were qualitatively measured using a SYBRGreen supermix (Bio-Rad Cat#1708880). Relative expression levels of the indicated genes were compared with GAPDH expression using the $2^{\Delta\Delta ct}$ method as previously described[34]. The primer sequences used for qRT-PCR were as listed in Supplementary Table 1.

**ChIP-seq, ChIP-qPCR, and SE definition**. ChIP-qPCR assays were performed as previously described[34]. Briefly, HNSCC cells were pretreated with 5 mM dimethyl 3,3'-dithiobispropionimidate-HCl (DTBP) (Thermo Fisher Scientific, cat#20665) and crosslinked with 1% formaldehyde. Chromosomes were broken into 200–500 bp genomic DNA fragments with a sonicator. Chromatin complexes were immunoprecipitated with the following antibodies: anti-p65 (Santa Cruz, cat#SC-372; 1:100); anti-MED1 (Bethyl, cat#A300-793A; 1: 200); anti-BRD4 (Abacm, cat#ab128874; 1:500); anti-H3K27Ac (Abcam, cat#ab4729; 1:500); anti-K3K4me1 (Abcam, cat#ab8895; 1:500); anti-CCNT1 (Bethyl, cat#A303-496A; 1:200); and anti-CDK9 (Bethyl, cat#A303-493A; 1:200) or control IgG. All precipitated DNA products were measured by qPCR. Data are shown as the percentage of input DNA. The primer sequences used for ChIP-qPCR were as listed in Supplementary Table 2.

ChIP-seq libraries were prepared with KAPA Hyper DNA Kit (KAPA Biosystems, Cat# KK8503). The workflow consisted of purification and fragmentation of ChIP DNA, end repair to generate blunt ends DNA, A-tailing, adaptor ligation and PCR amplification following the manufacturer's instruction. Different adaptors were used for multiplexing samples in one lane (KAPA Biosystems, KAPA Single-Indexed Adapter, Cat#KK8710). The samples were sequenced on Illumina HiSeq 3000 for a single read 50 run. Data quality check was examined on Illumina Sequencing Analysis Viewer (SAV). Demultiplexing was performed with Illumina Bcl2fastq2 v 2.17 program. Trimmomatic was used to remove adaptors and to trim quality bases. After adapter clipping, we removed leading and trailing ambiguous or low quality bases (below Phred quality scores of 3). The reads after quality trimming were aligned to the GRCh37(hg19) human reference genome using Bowtie. Only uniquely mapped reads were used for the downstream peak calling analysis. MACS2 was used for peak calling of the aligned data with default parameters ($P$-value $< 10^{-4}$) and the respective input samples were used as background. The bamcoverage utility in the deeptools was used to convert the bam to bw file for the IGV data visualization. R package chipseeker was used for the peak annotation. To designate ChIP-seq enriched regions (peaks) to genes, *Cis*-regulatory Elements Annotation System (CEAS) was used to generate average profiling of all Refseq genes and overlaps of significant peaks with genomic annotation regions as previously described[34].

SEs were identified using ROSE software Version 0.1 (https://bitbucket.org/young_computation/rose) as described before[3,4,10]. Briefly, the BRD4, MED1, or H3K27Ac peaks within 12.5 kb each another were stitched together as enhancer clusters, then ranked and plotted based on each ChIP-seq signal. Stitched enhancer clusters that pass the inflection point in the distribution were designated as SEs. Peaks were excluded if they were entirely contained within ±2 kb from a RefSeq

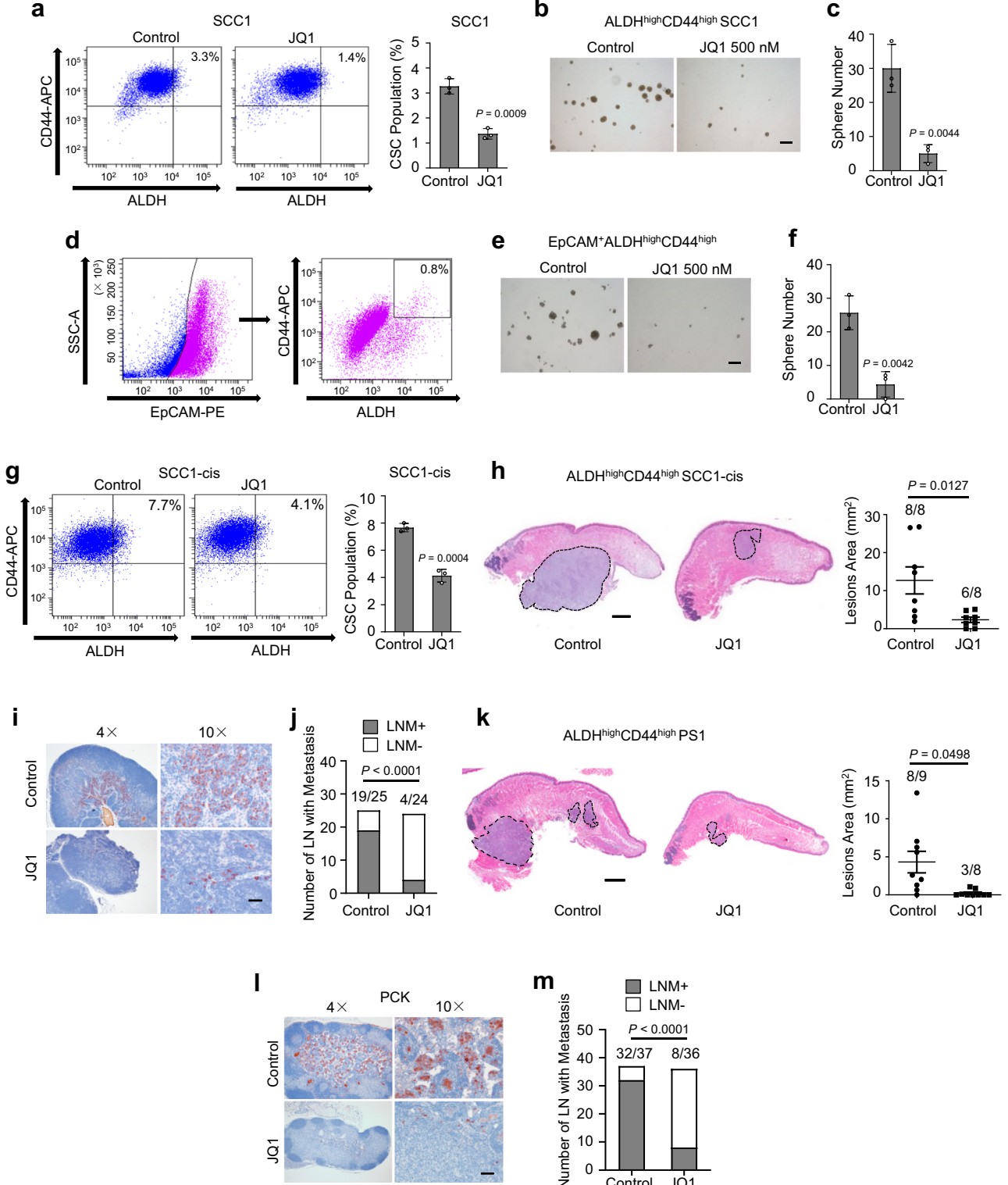

TSS. Enhancers were assigned to the RefSeq transcript whose TSS was nearest the center of the enhancer. The intersect utility in bedtools was used to calculate the overlapping of the SEs with other SEs or enhancers. The minimum overlap required is 1 bp.

**CRISPRa and CRISPRi assay**. For the CRISPRa assay, synergistic activation mediator (SAM) system was adopted[28]. Three lentivirus backbone plasmids that separately expressed dCas9-VP64 (VB170424-1009jup), MS2-P65-HSF1 (VB170424-1011udy), and msgRNAs (gRNAs with the MS2 loops, VB201112-1035tnv for MMP3 or VB201112-1038gcn for TP63) were constructed by VectorBuilder. For the CRISPRi assay, two lentiviral plasmids were constructed, one

expressing dCas9, KRAB, and MeCP2 (VB190101-1021vwr), another expressing the gRNAs (VB201112-1035tnv for MMP3 or VB201222-1082bxu for TP63). Lentivirus was generated and transduced into SCC1 cells. The transduced cells were selected with hygromycin, blasticidin, and puromycin. RNA was extracted and qRT-PCR was performed. The gRNA sequences for the MMP3 and TP63 enhancers were as listed in Supplementary Table 3.

**Luciferase reporter assay**. TP63-SE and TP63-NEG fragments were constructed into pLG4.23 luciferase reporter through standard PCR-clinging. The primer sequences for cloning TP63-SE were as listed in Supplementary Table 4. For luciferase assay, SCC1 cells were plated in 12-well plates at 40–50% confluence.

**Fig. 6 Disrupting SEs suppresses the tumorigenic potential and metastasis of CSCs from human HNSCCs. a** Representative FACS plots of CD44$^{high}$ALDH$^{high}$ cells in control and JQ1-treated SCC1 cells. Values are mean ± SD for triplicate experiments Statistical analysis was performed using two-tailed unpaired Student's *t*-test. **b** Representative images of tumorsphere formation assays for isolated CD44$^{high}$ALDH$^{high}$ cells upon JQ1 treatment. Scale bar, 200 μm. **c** Quantification of tumorspheres from isolated CD44$^{high}$ALDH$^{high}$ cells upon JQ1 treatment. Values are mean ± SD for triplicate experiments. Statistical analysis was performed using two-tailed unpaired Student's *t*-test. **d** Representative FACS plots of EpCAM$^+$CD44$^{high}$ALDH$^{high}$ CSCs from human PDXs of HNSCC. **e** Representative images of tumorsphere formation assays for isolated EpCAM$^+$CD44$^{high}$ALDH$^{high}$ CSCs upon JQ1 treatment. **f** Quantification of tumorspheres from EpCAM$^+$CD44$^{high}$ALDH$^{high}$ CSCs upon JQ1 treatment. Values are mean ± SD for triplicate experiments. Statistical analysis was performed using two-tailed unpaired Student's *t*-test. **g** Representative FACS plots and quantification of tumorspheres of CD44$^{high}$ALDH$^{high}$ CSC-like cells in control and JQ1-treated SCC1-cis cells. Values are mean ± SD for triplicate experiments. Statistical analysis was performed using two-tailed unpaired Student's *t*-test. **h** JQ1 inhibited orthotopic tumor growth of human CD44$^{high}$ALDH$^{high}$ CSC-like cells from SCC1-cis cells ($n = 8$ per group). Statistical analysis was performed using two-tailed unpaired Student's *t*-test. Scale bar, 1 mm. **i** Immunostaining of lymph node metastasis with anti-pan cytokeratin antibodies ($n = 8$ per group; 25 lymph nodes were retrieved from Control group, and 24 lymph nodes were retrieved from JQ1 treatment group). Scale bar, 50 μm. **j** JQ1 inhibited lymph node metastasis of human CD44$^{high}$ALDH$^{high}$ CSC-like cells from SCC1-cis cells ($n = 8$ per group; 25 lymph nodes were retrieved from Control group, and 24 lymph nodes were retrieved from JQ1 treatment group). Statistical analysis was performed using two-tailed Fisher's exact test. **k** JQ1 inhibited orthotopic tumor growth of EpCAM$^+$CD44$^{high}$ALDH$^{high}$ CSCs from human PDXs of HNSCC ($n = 9$ for Control group; $n = 8$ for JQ1 treatment group). Statistical analysis was performed using two-tailed Student's *t*-test. Scale bar, 1 mm. **l** Immunostaining of metastatic tumor cells in cervical lymph nodes using anti-pan-keratin ($n = 9$ for Control group, 37 lymph nodes were retrieved from this group; $n = 8$ for JQ1 treatment group, 36 lymph nodes were retrieved from this group). **m** JQ1 inhibited lymph node metastasis of human EpCAM$^+$CD44$^{high}$ALDH$^{high}$ CSCs from human HNSCC samples ($n = 9$ for Control group, 37 lymph nodes were retrieved from this group; $n = 8$ for JQ1 treatment group, 36 lymph nodes were retrieved from this group). Statistical analysis was performed using two-tailed Fisher's exact test. Scale bar, 50 μm. The experiments in **h**–**m** were repeated once with similar results.

After 12 h, the cells were transfected with 100 ng of luciferase reporters and 50 ng of CMV-β-galactosidase constructs using Lipofectamine 2000. Two days post transfection, the luciferase and β-galactosidase activity of total cell lysates were determined by Bright-Glo™ Luciferase Assay System (Promega; cat#E2620) and GalactoStar Reporter Gene Assay Stystem (Applied Biosystems, cat#T1012). The luciferase reporter activity was normalized against the β-galactosidase activity of each cell lysate sample.

**Statistical analysis**. Numerical data and histograms were expressed as the mean ± SD. Two-tailed Student's *t*-test or the nonparametric Mann–Whitney test was performed between two groups, and $P < 0.05$ was considered statistically significant. For percentage of mice with positive lymph node metastasis, Fisher's exact test was performed. Other statistical methods were specifically indicated. All statistical analyses were performed with GraphPad Prism software (Version 8.2.0.).

## Data availability

The authors declare that all relevant data are available within the article and its supplementary information files or from the corresponding authors upon reasonable request. The RNA-seq and ChIP-seq datasets generated in the course of this study are available on the NCBI GEO database under the accession numbers GSE131967 and GSE131710, respectively. Source data are provided with this paper.

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

## Acknowledgements

This work was supported by National Institute of Health grants R01DE029173 and R03DE026822.

## Author contributions

The study was conceived and designed by J.L. and C.W. J.D. and J.L. performed all experiments. ChIP-seq data analysis was performed by Y.Y., and J.L. and Y.L. assisted J.D. for animal studies and histology. The manuscript was written by J.D., J.L., and C.Y.W. with input with other authors.

## Competing interests

The authors declare no competing interests.
