## [Peer Review File · Nature Communications]

REVIEWER COMMENTS

Reviewer #1 (Remarks to the Author):

In this manuscript, the authors demonstrated that super-enhancers (SEs) transcriptionally regulated stemness and metastatic gene expression in HNSCC, thereby controlling tumorigenic potential and metastasis. They found that BRD4 recruited MED1 and p65 to the SEs of TP63 and MET, which are critical genes involved in stemness maintenance. Both in vitro and in vivo experiments revealed that disrupting SEs by BET inhibitors potentially inhibited self-renewal CSC of HNSCC and eliminated CSCs in addition to the elimination of proliferating non-stem tumor cells. The study is potentially interesting; however, there are several concerns need to be addressed.

1. The interaction between Brd4 and NF- κ B signaling pathways has been published in other cancer cell lines (PMID: 19103749,23686307,19103749,24189064), which reduces the overall novelty of this study.
2. The authors observed the dysregulation of genes within NF- κ B signaling pathways in JQ1 treated cells. It is unclear whether the dysregulation is due to the alteration of gene expression of NF- κ B components, such as RelA/B, I κ B α . Real-time qPCR and western blot analysis could address this concern.
3. The entire manuscript is heavily dependent on the sequencing analysis, including RNA-seq and ChIP-seq. Although the authors provided the GEO access information of related sequencing data. It would be more helpful to provide supplementary tables to provide detailed information related to sequencing data information. For example, the list of differentially expressed genes and more detailed information on GSEA analysis.
4. Figure 1F and G showed that JQ1 down-regulated stemness and NF- κ B targeted genes in CSC. Will CSC and non-CSC exhibit different drug response to JQ1 treatment? It would be more convincing to include non-ALDH^{high}CD44^{high} population as control to clarify this point.
5. Among all the differentially expressed genes (including up- and down-regulated genes) identified between the control and JQ1 treated groups, how many genes with altered expression could be explained by JQ1 mediated SE interruption? Does the JQ1 mediated SE interruption could explain the majority changes in the transcriptional outputs? If not, it would be more helpful to discuss the potential mechanism that could be involved in regulating non-SE regulated transcriptional regulation by JQ1 treatment.
6. The authors observed dramatic genome-wide decrease of p65 binding in JQ1 treated cells compared with the control group. While the genome-wide decrease of BRD4 is not as dramatic as p65 (Figure 2e-f). On the other hand, the authors claimed that p65 binding depends on BRD4 based on data illustrated in Figure 4. It is unclear whether the observed BRD4 dependent p65 binding is limited to tested loci, or is this a general mechanism. A genome-wide ChIP-seq analysis of p65 and BRD4 in BRD4 KD and p65 KD cells, respectively, could clarify this important point.
7. The authors claimed that the genomic regions illustrated in Figure 3C, E, G, I, are super-enhancers to their corresponding genes. However, such claim is based on the genomic location, but not functional annotation. To further confirm that these regions are indeed the super enhancers to their corresponding genes, CRISPR based the genome editing or epigenome editing is needed to clarify this critical point.
8. The authors observed that genomic binding of p65 or MED1 is dependent on BRD4. However, the underlying mechanism is unclear. Previous studies reported the direct interaction between BRD4 and p65 (PMID: 9103749,23686307,19103749,24189064). It is unclear whether these three components directly interact with each other in this case; or other factors involved in the interaction? A more detailed molecular analysis, e.g., Co-IP, could be used to clarify this point.
9. In Figure 4H, the authors observed higher TP63 and MET gene expression in ALDH^{high} CD44^{high} CSCs than ALDH^{low}CD44^{high} population. However, it is unclear whether such differential expression is due to the

dysregulation of enhancers. A genome-wide or loci-specific analysis of H3K27ac, Brd4 and p65 binding in these two cell types are suggested.

10. A rescue experiment by overexpression of TP63 or MET in HNSCC cells treated with/without JQ1 should be performed to confirm that they are critical downstream effectors in terms of stemness of tumor cells.

11. The author observed decreased CD44^{high}ALDH^{high} cells in the JQ1 treated group using FACS analysis. It would be more convincing to add statistical quantifications of FACS data since the changes seemed to be relatively minor based on the FACS plot shown in Figure 6a, d, g.

12. The observation that JQ1 treatment overcomes cisplatin resistance is interesting; however, based on the data shown in Figure 6, the underlying mechanism is unclear. Is that because these two drugs targeting different cell types (CSC and non-CSC), or targeting different regulatory pathways in the same cell population (DNA damage vs enhancer related transcription regulation). A more detailed analysis need to be performed to clarify this.

Reviewer #2 (Remarks to the Author):

In this manuscript, the authors identified the roles of transcriptional super-enhancers in control of cancer stemness and metastasis-associated gene expression in squamous cell carcinomas. They identified target genes and further validated functional contributions of super-enhancers using BET inhibitor. Overall, the study is novel and identified significant molecular mechanisms of super-enhancers in cancer. I have the following minor concerns:

- 1) Title: "Super-Enhancers" should be explicit as "transcriptional super-enhancers". "control cancer stemness and metastasis" is overstated without directly taking out super-enhance element to assess the impact on stemness and metastasis. It would be precise as "control cancer stemness and metastasis-associated genes", which the authors provided direct evidence.
- 2) Abstract: should define p65 as NFkB-p65, and also bunch of gene names were used without defining them or specifying they are gene names, e.g., BRD4, MED1, BET, which makes the abstract hard to read for readers outside of the field.
- 3) Introduction: the authors stated "MYC is not a dominant oncogene" in human HNSCC, this may be overstated. MYC may not be a main BET target, it has complex protein level regulation to determine its stability and activity.
- 4) The functional impact of super-enhances on tumor growth and metastasis would be strengthened if authors could perform in vivo ChIP-seq to show JQ1 treated tumors indeed have reduced binding of regulators to super enhancers. It would be laborious and challenging with limited tumor materials to examine all of the regulators in parallel with Fig. 3, even with one or two would be nice but I understand the difficulties.
- 5) The authors may add discussion for the translational impact of this work, e.g., do they see BET inhibitors could move to clinic in treating head and neck cancer?

Xiao-Jing Wang

Reviewer #3 (Remarks to the Author):

In the manuscript by Dong and colleagues the authors demonstrate that BET inhibitors (JQ-1 and I-BET-151) significantly decrease the expression of key genes required to maintain cancer stem-like and pro-metastatic phenotypes using established HNSCC cell lines, lineage tracing HNSCC mouse model, and patient-derived xenograft models. Unlike previous reports in the literature where BET inhibitors were used to

target MYC-driven tumor types, evidence provided in this body of work indicates that JQ1 and I-BET-151 impact HNSCC gene expression from super-enhancers largely through MYC-independent mechanisms. Using ChIP-seq, ChIP-qPCR, and siRNA knockdown experiments the authors demonstrate that in HNSCC super-enhancers are composed of BRD4, MED1, and the p65 subunit of NF-kappaB. Evidence provided indicate that BRD4 recruits MED1 and p65 to super-enhancer structures, which in turn are required to maintain cancer stem-like properties and govern tumorigenic potential. Using a HNSCC mouse model, evidence is provided that BETi therapy decrease lymph node metastases, in part, by eliminating cancer stem-like and non-stem-like HNSCC cell populations. Lastly, the authors demonstrated that BET inhibition also reverses pro-tumor phenotypes in chemoresistant HNSCC cell lines, indicating that BETi therapy could potentially be used as a secondary line of treatment for recurrent disease. Overall, the research findings presented in this study are of the highest importance, and will undoubtedly have a long-lasting impact to the field of chromatin-cancer biology.

Despite the impact of the scientific findings by Dong and colleagues, minor criticisms regarding the manuscript need to be addressed.

Concerns:

1. The authors fail to mention in the Discussion that BETi treatment impacts cancer stem-like properties in chemoresistant cells. Given the clinical implications, this point warrants discussion.

Minor criticisms:

1. Minor grammatical errors were noted throughout the manuscript.

2. Inaccurate primary figure panels and supplemental figures were misreferenced in the manuscript text on multiple occasions.

3. Statistical tests were missing from the figure legends. This information would have been helpful when interpreting Figure 1E Fadu panel where the bar grafts are denoted with **, yet exhibit rather large standard deviations.

4. Figure 2G is not described in the body of the manuscript.

5. The scale bar on the histopathological slides does not indicate the magnification.

6. In general, specific experimental details regarding methods are missing. For example, how siRNA knockdowns were performed and the number of experimental replicates for all experiments are poorly described.

We would like to thank the reviewers for carefully reading our manuscript and for their constructive comments which have further helped us improve our manuscript. The following is our point by point response to the reviewers' comments.

Reviewer #1

1. The Interaction between Brd4 and NF- κ B signaling pathways has been published in other cancer cell lines (PMID: 19103749,23686307,19103749,24189064), which reduces the overall novelty of this study.

Indeed, the interaction between BRD4 and NF- κ B has been mentioned in other studies. However, we feel that the findings presented in our manuscript are novel based on the following reasons: 1) We showed that NF- κ B p65 selectively associates with BRD4 and MED1 to establish SEs at key cancer stemness and pro-metastatic genes to maintain the functional properties of CSCs in HNSCC; 2) Using in vivo lineage tracing, we demonstrated that targeting SEs potently inhibited tumor metastasis by eliminating cancer stem cells; and 3) We revealed that the recruitment of p65 to promoters and enhancers, especially SEs, is dependent on BRD4 in HNSCC.

2. The authors observed the dysregulation of genes within NF- κ B signaling pathways in JQ1 treated cells. It is unclear whether the dysregulation is due to the alteration of gene expression of NF- κ B components, such as RelA/B, I κ B α . Real-time qPCR and western blot analysis could address this concern.

We performed additional qRT-PCR analysis and found JQ1 did not affect the expression of *RELA*, *RELB*, and *IKBA* in SCC cells (**Supplementary Fig. 1a**).

3. The entire manuscript is heavily dependent on the sequencing analysis, including RNA-seq and ChIP-seq. Although the authors provided the GEO access information of related sequencing data. It would be more helpful to provide supplementary tables to provide detailed information related to sequencing data information. For example, the list of differentially expressed genes and more detailed information on GSEA analysis.

We have included the list of differentially expressed genes in SCC1, SCC22B, and FaDu cells after JQ1 treatment in the supplementary materials. In addition, we also included the gene list of the NF- κ B and Cancer Stemness for GSEA.

4. Figure 1F and G showed that JQ1 down-regulated stemness and NF- κ B targeted genes in CSC. Will CSC and non-CSC exhibit different drug response to JQ1 treatment? It would be more convincing to include non-ALDH^{high}CD44^{high} population as control to clarify this point.

We isolated both CSC and non-CSCs and performed the JQ1 treatment. JQ1 had little effect on *TP63* and *MET* expression in non-CSCs compared with CSCs (**Supplementary Fig. 4e**).

5. Among all the differentially expressed genes (including up- and down-regulated genes) identified between the control and JQ1 treated groups, how many genes with altered expression could be explained by JQ1 mediated SE interruption? Does the JQ1 mediated SE interruption could explain the majority changes in the transcriptional outputs? If not, it would be more helpful to discuss the potential mechanism that could be involved in regulating non-SE regulated transcriptional regulation by JQ1 treatment.

We did the analysis as Reviewer 1 suggested. Among all 3648 differentially expressed genes regulated after JQ1 treatment, we identified 227 up-regulated genes and 193-down regulated genes that are associated with SEs. Thus, JQ1-mediated SE interruption does not represent the majority changes in the transcriptional output. Studies have shown that key oncogenes driven by

SEs are vulnerable to SE disruption¹⁻³. Therefore, we asked whether SE-associated transcripts are more disproportionately relying on SEs than typical enhancers (TEs). An analysis of transcriptional profiles indicated that SE-associated transcripts were significantly more expressed than the TE-associated transcripts (**Fig. 2d**). In addition, among the genes downregulated by JQ1, the abundance of SE-associated transcripts was downregulated to a significantly higher degree upon JQ1 treatment as compared with those associated with TEs (**Fig. 2e**), thus indicating that the high expression of SE-associated genes are, in particular, transcriptionally associated with SEs.

6. The authors observed dramatic genome-wide decrease of p65 binding in JQ1 treated cells compared with the control group. While the genome-wide decrease of BRD4 is not as dramatic as p65 (Figure 2e-f). On the other hand, the authors claimed that p65 binding depends on BRD4 based on data illustrated in Figure 4. It is unclear whether the observed BRD4 dependent p65 binding is limited to tested loci, or is this a general mechanism. A genome-wide ChIP-seq analysis of p65 and BRD4 in BRD4 KD and p65 KD cells, respectively, could clarify this important point.

We have demonstrated that the recruitment of p65 to SE regions is BRD4 dependent. To explore whether such BRD4-dependent p65 recruitment is also extended to other loci besides SE regions, such as promoters or enhancers, we selected several loci and performed the ChIP-qPCR analysis. These regions include the promoters of *CXCL1* and *ICAM1* and the enhancer region of *LBT*. These are well-characterized NF- κ B target genes. Based on our ChIP-seq data, both BRD4 and p65 are recruited to these sites (**Supplementary Fig. 4c**). Similar to the findings at SE regions, ChIP-qPCR analysis indicated that depletion of p65 has little effect on BRD4 recruitment (**Supplementary Fig. 4c**). In comparison, significant loss of p65 recruitment to these regions has been found in BRD4 depleted SCC1 cells (**Supplementary Fig. 4c**). Collectively, these results suggest that the recruitment of p65 to chromatin by BRD4 is a general and also a critical transcriptional mechanism in SCC.

Previous reports indicated that the acetylation lysine is required for the interaction of p65 with BRD4 in 293T cells⁴. JQ1 may block the interaction between BRD4 with p65. To test this possibility, we performed the *in vitro* GST-pull down assays. As show in **Supplementary Fig. 2d**, GST-BRD4-49-460, which contains both BD1 and BD2 bromodomains, pulled down overexpressed Flag-tagged p65 in 293T cell lysate. JQ1 treatment inhibited their interaction in a dose dependent manner. Taken together, our results strongly suggest that the genome-wide p65 recruitment is BRD4 dependent.

7. The authors claimed that the genomic regions illustrated in Figure 3C, E, G, I, are super-enhancers to their corresponding genes. However, such claim is based on the genomic location, but not functional annotation. To further confirm that these regions are indeed the super enhancers to their corresponding genes, CRISPR based the genome editing or epigenome editing is needed to clarify this critical point.

This is an excellent suggestion. We adopted the CRISPRa and CRISPRi techniques to recruit either a transcriptional activator or repressor complex to the SE loci with a catalytically dead Cas9 (dCas9) and examine the role of SEs in regulating *TP63* and *MMP3* expression. As shown in **Fig. 3k** and **3l**, the recruitment of CRISPR Synergistic Activation Mediator (SAM) complex by CRISPRa to both *TP63* SEs regions resulted in significant upregulation of *TP63* expression in SCC1 cells. In comparison, tethered dCas9-KRAB repressor to the same loci by CRISPRi led to significant inhibition of *TP63*. We also obtained a similar result on *MMP3*-SE. To further confirm these findings, we cloned a 2 kb fragment of *TP63* SEs region as well as a 2 kb negative control region (*TP63*-NEG) into the pLG4.23 luciferase reporter (**Fig. 3m**). As compared to the negative

control and pLG4.23 promoter, the TP63 SE fragment was capable of elevating the luciferase reporter activity in SCC1 cells (**Fig. 3m**). Taken together, these results confirmed the functional regulatory role of SE in transcriptional activation.

8. The authors observed that genomic binding of p65 or MED1 is dependent on BRD4. However, the underlying mechanism is unclear. Previous studies reported the direct interaction between BRD4 and p65 (PMID: 9103749,23686307,19103749,24189064). It is unclear whether these three components directly interact with each other in this case; or other factors involved in the interaction? A more detailed molecular analysis, e.g., Co-IP, could be used to clarify this point.

As the Reviewer suggested, we performed the endogenous IP and confirmed the interaction between BRD4 and p65 in SCC1 cells (**Supplementary Fig. 2c**). We also found MED1 can also interact with p65 endogenously (**Supplementary Fig. 2c**). Additionally, we confirmed the interaction between p65 and MED1 was BRD4-dependent in SCC1 cells by knocking down BRD4 (**Supplementary Fig. 4b**). A previous report indicated p65 binds to BRD4 directly, and the acetylation lysine is required for the interaction of p65 with BRD4 in 293T cells⁴. To test whether JQ1 may directly block the interaction between BRD4 and p65, we performed an *in vitro* GST-pull down assays. As show in **Supplementary Fig. 2d**, GST-BRD4-49-460, which contains both BD1 and BD2 domains, can pull down overexpressed Flag-tagged p65 in 293T cell lysate. JQ1 inhibited their interaction in a dose dependent manner. Taken together, our results further confirmed that the genomic binding of p65 or MED1 is dependent on BRD4 and can be suppressed by JQ1 directly.

9. In Figure 4H, the authors observed higher TP63 and MET gene expression in ALDH^{high} CD44^{high} CSCs than ALDH^{low}CD44^{high} population. However, it is unclear whether such differential expression is due to the dysregulation of enhancers. A genome-wide or loci-specific analysis of H3K27ac, Brd4 and p65 binding in these two cell types are suggested.

CSCs only represent a small population, around 3% in SCC1 cells (**Fig. 6a**). ChIP-seq normally requires large amount of cells to start with, thus, technically it would be impractical to perform ChIP-seq using ALDH^{high} CD44^{high} CSCs. However, we have included the ChIP-qPCR analysis of these two different population of cells. Increased enrichments of both p65 and BRD4 on SEs were observed in CSCs as compared with ALDH^{low} CD44^{high} non-stem tumor cells.

10. A rescue experiment by overexpression of TP63 or MET in HNSCC cells treated with/without JQ1 should be performed to confirm that they are critical downstream effectors in terms of stemness of tumor cells.

We performed the rescue experiments by overexpressing MET in SCC1 cells (**Supplementary Fig. 3a**). While over-expression of MET in SCC1 cells significantly increased tumorsphere formation, the inhibition of tumorsphere formation by JQ1 was less potent in SCC1 cells overexpressing MET than in SCC1 cells expressing empty vector (**Supplementary Fig. 3b**). Similarly, overexpression of MET could also recue the inhibition of SCC1 cell migration by JQ1 (**Supplementary Fig. 3c**). Of note, overexpression of MET could not fully rescue JQ1 inhibition due to the fact that JQ1 probably inhibited the expression of other functional genes associated with SCC.

11. The author observed decreased CD44^{high}ALDH^{high} cells in the JQ1 treated group using FACS analysis. It would be more convincing to add statistical quantifications of FACS data since the changes seemed to be relatively minor based on the FACS plot shown in Figure 6a, d, g.

We have included the statistical analysis in the revised manuscript. Please see **Fig. 6a, Fig. 6g**, and **Supplementary Fig. 5a**.

12. The observation that JQ1 treatment overcomes cisplatin resistance is interesting; however, based on the data shown in Figure 6, the underlying mechanism is unclear. Is that because these two drugs targeting different cell types (CSC and non-CSC), or targeting different regulatory pathways in the same cell population (DNA damage vs enhancer related transcription regulation). A more detailed analysis need to be performed to clarify this.

Our previous studies indicate that increased FOSL1 activity facilitates metastasis and cisplatin resistance of Bmi1⁺CSC in a spontaneous mouse model of HNSCC⁵. Coinciding with this finding, similar phenomena has also been found in human HNSCC cells^{5, 6}. As shown in **Supplementary Fig. 5e**, upregulation of *BMI1* and *FOSL1* has been found in SCC1-cis cells. Such upregulation can be suppressed by JQ1. This indicates that JQ1 may overcome cisplatin resistance through the inhibition of *BMI1* and *FOSL1* expression. Besides these two cancer stemness genes, the key oncogenes associated with cell proliferation, such as *CCND1*, can also be significantly suppressed by JQ1 (**Supplementary Fig. 5e**). Taken together, our results indicate that JQ1 can target different sub-population or phenotypes of HNSCC cells (cancer stemness and fast proliferation) through the inhibition of different sets of genes.

Reviewer #2:

1) Title: “Super-Enhancers” should be explicit as “transcriptional super-enhancers”. “control cancer stemness and metastasis” is overstated without directly taking out super-enhance element to assess the impact on stemness and metastasis. It would be precise as “control cancer stemness and metastasis-associated genes”, which the authors provided direct evidence.

We appreciate this comment. We have added the transcriptional” in our title and revised our statement as suggested.

2) Abstract: should define p65 as NFkB-p65, and also bunch of gene names were used without defining them or specifying they are gene names, e.g., BRD4, MED1, BET, which makes the abstract hard to read for readers outside of the field.

We have defined “p65” as NF-κB-p65. Bromodomain-containing protein 4 is fully spelled out and defined as BRD4 in the abstract. We have defined MED1, BET and other gene names.

3) Introduction: the authors stated “MYC is not a dominant oncogene” in human HNSCC, this may be overstated. MYC may not be a main BET target, it has complex protein level regulation to determine its stability and activity.

We agree with Reviewer’s comments. We have deleted our statement in the introduction accordingly.

4) The functional impact of super-enhances on tumor growth and metastasis would be strengthened if authors could perform in vivo ChIP-seq to show JQ1 treated tumors indeed have reduced binding of regulators to super enhancers. It would be laborious and challenging with limited tumor materials to examine all of the regulators in parallel with Fig. 3, even with one or two would be nice but I understand the difficulties.

This is an excellent suggestion. However, due to the current COVID-19 pandemic, the animal facility in our institution has been maintained at husbandry level. We have very limited access for the animal facilities and animal ordering to perform any animal related studies in a timely fashion. In addition, ChIP-seq usually requires large amount of cells to start with. Based on our experience,

the yield of isolation of tumor cells from HNSCC PDXs is extremely low as compared to other type of tumors. Thus, the suggested experiments would need large quantity of mice. Based on these restrictions, unfortunately it is not realistic for us to perform the suggested studies. However, to further demonstrate that the functional role of SEs in SCC, we adopted the CRISPRa and CRISPRi techniques to recruit either a transcriptional activator or repressor complex to the SE loci with a catalytically dead Cas9 (dCas9) and examine the role of SEs in regulating *TP63* and *MMP3* expression. We confirmed the functional regulatory role of SE in transcriptional activation in SCCs (**Fig. 3k** and **3l**; Also see Reviewer 1, point 7).

5) The authors may add discussion for the translational impact of this work, e.g., do they see BET inhibitors could move to clinic in treating head and neck cancer?

We have discussed the translational value of our work in the discussion section as the Reviewer suggested.

Reviewer #3

1. The authors fail to mention in the Discussion that BETi treatment impacts cancer stem-like properties in chemoresistant cells. Given the clinical implications, this point warrants discussion.

We appreciate this comment, and we have discussed the therapeutic potential of BET inhibition in overcoming drug resistance in the discussion section in our revised manuscript.

Minor criticisms:

1. Minor grammatical errors were noted throughout the manuscript.

We have carefully edited our manuscript and corrected these errors.

2. Inaccurate primary figure panels and supplemental figures were misreferenced in the manuscript text on multiple occasions.

We corrected these errors.

3. Statistical tests were missing from the figure legends. This information would have been helpful when interpreting Figure 1E Fadu panel where the bar graphs are denoted with **, yet exhibit rather large standard deviations.

We have included statistic information in each figure.

4. Figure 2G is not described in the body of the manuscript.

We have corrected this typo.

5. The scale bar on the histopathological slides does not indicate the magnification.

We have included the information for the related figures in our revised manuscript.

6. In general, specific experimental details regarding methods are missing. For example, how siRNA knockdowns were performed and the number of experimental replicates for all experiments are poorly described.

We have added the specific experimental details in the section of Materials and Methods in our revised manuscript.

References:

1. Hnisz, D. *et al.* Super-enhancers in the control of cell identity and disease. *Cell* **155**, 934-947 (2013).
2. Loven, J. *et al.* Selective inhibition of tumor oncogenes by disruption of super-enhancers. *Cell* **153**, 320-334 (2013).
3. Zanconato, F. *et al.* Transcriptional addiction in cancer cells is mediated by YAP/TAZ through BRD4. *Nat Med* **24**, 1599-1610 (2018).
4. Huang, B., Yang, X.D., Zhou, M.M., Ozato, K. & Chen, L.F. Brd4 coactivates transcriptional activation of NF-kappaB via specific binding to acetylated RelA. *Mol Cell Biol* **29**, 1375-1387 (2009).
5. Chen, D. *et al.* Targeting BMI1(+) Cancer Stem Cells Overcomes Chemoresistance and Inhibits Metastases in Squamous Cell Carcinoma. *Cell Stem Cell* **20**, 621-634 e626 (2017).
6. Nor, C. *et al.* Cisplatin induces Bmi-1 and enhances the stem cell fraction in head and neck cancer. *Neoplasia* **16**, 137-146 (2014).

REVIEWERS' COMMENTS

Reviewer #1 (Remarks to the Author):

The authors addressed all comments.

Reviewer #2 (Remarks to the Author):

The authors made serious effort to address reviewers' concerns experimentally. I have no additional concerns.

Reviewer #3 (Remarks to the Author):

In this revised manuscript by Dong and colleagues, the authors provide compelling evidence that NF-kappaB is critical for super-enhancer activity by the ability of RelA/p65 to physically recruit BRD4 and MED1. Experimental evidence is provided that this chromatin-associated complex is essential to drive the expression of genes required to establish cancer stem-cell phenotypes and promote dedifferentiated transcriptional programs that promote tumor metastasis in head and neck squamous cell carcinomas. The authors have been extremely responsive to the previous reviewer's comments, providing additional experimental evidence that further supports their original hypothesis. This is an impressive body of work that will have broad and long-standing impact in the fields of cancer genomics and cancer stem-cell biology.

The reviewers are satisfied with our responses. There are no additional concerns.